# ET-Plan-Bench: Embodied Task-level Planning Benchmark Towards Spatial-Temporal Cognition with Foundation Models

## Abstract

Recent advancements in Large Language Models (LLMs) have spurred numerous attempts to apply these technologies to embodied tasks, particularly focusing on high-level task planning and task decomposition. To further explore this area, we introduce a new embodied task planning benchmark, ET-Plan-Bench [1], which specifically targets embodied task planning using LLMs. It features a controllable and diverse set of embodied tasks varying in different levels of difficulties and complexities, and is designed to evaluate two critical dimensions of LLMs' application in embodied task understanding: spatial (relation constraint, occlusion for target objects) and temporal & causal understanding of the sequence of actions in the environment. By using multi-source simulators as the backend simulator, it can provide immediate environment feedback to LLMs, which enables LLMs to interact dynamically with the environment and re-plan as necessary. We evaluated the state-of-the-art open source and closed source foundation models, including GPT-4, LLAMA and Mistral on our proposed benchmark. While they perform adequately well on simple navigation tasks, their performance can significantly deteriorate when faced with tasks that require a deeper understanding of spatial, temporal, and causal relationships. Thus, our benchmark distinguishes itself as a large-scale, quantifiable, highly automated, and fine-grained diagnostic framework that presents a significant challenge to the latest foundation models. We hope it can spark and drive further research in embodied task planning using foundation models.

## 1 Introduction

Embodied planning refers to an agent's ability to formulate plans and execute tasks within a physical environment. Large Language Models (LLMs) and Vision Language Models (VLMs) have demonstrated significant advancements in vision understanding, natural language comprehension, and generation. Although LLMs and VLMs are not inherently designed for embodied planning, there is potential for these models to contribute to this field. LLMs and VLMs possess an extensive repository of knowledge derived from their training data, which enables them to comprehend and generate contextually relevant advice and strategies. Additionally, they have the capability to translate complex tasks into step-by-step instructions through interactions with humans. Furthermore, they can refine plans based on feedback from the environment or human interventions. In the context of embodied planning, LLMs focus on the decomposition of complex tasks. The LLM acts both as a translator and a mediator. Each sub-task, such as navigating to a specific location or grabbing a particular object, can be executed using advanced Reinforcement Learning (RL) methods. However, currently, LLMs and VLMs face some challenges in understanding the physical world (Jia et al., 2022; Chen et al., 2024; Jain et al., 2023), including spatial understanding. Effective spatiotemporal reasoning often necessitates the integration of knowledge from multiple domains, such as physics and human behavior. These domains may not be adequately represented in the training data of existing foundational models. To further explore this area, we introduce a new embodied task planning benchmark, *ET-Plan-Bench*,

---

[1]The benchmark and source code will be publicly released once the paper is accepted. Attached in the supplementary material for this submission.

which features an automatic embodied task generation and evaluation pipeline that is designed to evaluate tasks with *spatial* and *temporal* understanding of the environment.

For *spatial* understanding, our benchmark aims to assess the LLMs' ability to complete tasks with relational, size, and occlusion constraints between objects, which are pivotal for effective interaction within a given space. In our benchmark, the spatial aspect is currently not associated with the understanding of left and right directions. In terms of *temporal* understanding, we focus on evaluating LLMs' ability to understand the causality and sequence between preceding and subsequent actions during the execution of embodied tasks. This aspect tests how well the models can plan and execute actions that depend on each other in a meaningful sequence.

Compared to existing benchmarks in the field of embodied AI, our approach offers unique features that allows a thorough evaluation on existing foundation models for their embodied task planning ability. Distinguishing itself from the standard benchmarks that often rely on manual curation for embodied tasks (Puig et al., 2018; 2020; Shridhar et al., 2020a; Li et al., 2023; Wan et al., 2022), our framework employs LLMs to autonomously generate a wide array of embodied tasks with different spatial and temporal constraints in a systematic way, incorporating tasks with variety of objects, scene and room layouts in different level of difficulties. Another important category of benchmarks is embodied question answering (EQA), which is designed to evaluate the capability of LLMs to answer questions based on first-person or third-person videos instead of task decomposition and long-horizon planning (Jia et al., 2022; Majumdar et al., 2024; Cheng et al., 2024). However, most of these EQA datasets depend on videos and lack interaction with the physical or vitual environment, while our benchmark features end-to-end (long-horizon) task planning through interaction with a dynamic environment.

This comprehensive embodied benchmark aims to identify both the capabilities and limitations of current foundation models in navigating and interacting within complex, dynamic environments. Our findings are intended to provide deeper insights into the practical applications of these models in real-world settings. The main contributions of our work state as follows:

**Embodied task generation with various difficulties** We have developed an embodied task generation pipeline which can produce different levels of difficulties and complexities through introducing *spatial* and *temporal* constraints. It supports automatic task generation and task success criteria generation, allowing automatic task planning evaluation in an end-to-end manner.

**Benchmark:** Using the task generation pipeline, we introduce a benchmark focusing on various aspects of spatial (relation constrain, object occlusion and global layout map) and temporal understanding (actions dependency and optimal moving path for robot). It provides a detailed and thorough diagnostic assessment of existing foundation models.

**LLM agent baseline comparison:** We provide a comprehensive set of our proposed LLM agent baseline results, featuring both recently released LLMs such as GPT-4 and open-source foundation model like LLAMA (Touvron et al., 2023) and Mixtral 8x7B (Jiang et al., 2024). Particularly, we demonstrates that a supervised fine-tuned (SFT) small-size LLAMA-7B model using our benchmark EQA data can match the state-of-the-art closed-source model GPT-4, highlighting the effectiveness of our benchmark in enhancing understanding of embodied environments.

## 2 RELATED WORK

Table 1 presents a comparison of recent benchmarks for household embodied planning. This comparison spans multiple dimensions, including modality (natural language, vision, or both), vocabulary type (open or constrained), data size, and the capacity for unlimited data generation via the provided pipeline. Additionally, it assesses whether benchmarks evaluate the planning abilities of large language models (LLMs), whether data generation is automated or human-annotated, and the consideration of spatial, temporal, or causal constraints. We provide a concise survey of related work here; a comprehensive version is available in the Appendix A.3.

**Evaluation of foundation models for household embodied planning** LLMs are used in task planning for their generalization capabilities across various tasks(Brown et al., 2020; Ahn et al., 2022; Huang et al., 2022). Recent work shows significant advances in using LLMs for long-horizon planning (Zhao et al., 2024), but challenges like hallucination and poor spatial reasoning remain

Table 1: Comparison of our proposed dataset with prior work. **EQA**: Embodied Question Answering, **MQA**: Multiple Question Answering, **TGP**: Task Generation & Planning, **P**: Partial, **G**: Global, **FP**: Partial observation with large Furnitures

| Dataset | Task | Multi-Modality | Data Size | Auto Data | LLM Eval | Open Voc | Level of Obs | Spatial | Temporal/Causal | Env Inter |
|---|---|---|---|---|---|---|---|---|---|---|
| ActivityPrograms(Puig et al., 2018) | | ✔ | 2821 | ✗ | ✗ | ✔ | P | ✗ | ✗ | ✔ |
| WAH(Puig et al., 2020) | | ✗ | 1211 | ✗ | ✗ | ✗ | P/ G | ✗ | ✗ | ✔ |
| ALFRED(Shridhar et al., 2020a) | | ✔ | 8055 | ✗ | ✗ | ✔ | P | ✗ | ✗ | ✔ |
| WAH-NL(Choi et al., 2024) | | ✗ | 611 | ✗ | ✔ | ✔ | P | ✗ | ✗ | ✔ |
| RoboGen(Wang et al., 2023) | | ✔ | ∞ | ✔ | ✔ | ✔ | P | ✗ | ✗ | ✔ |
| BEHAVIOR(Srivastava et al., 2022) | | ✗ | 100 | ✗ | ✗ | ✗ | P | ✗ | ✗ | ✔ |
| Mini-BEHAVIOR(Jin et al., 2023) | TGP | ✗ | 20 | ✗ | ✗ | ✗ | P | ✗ | ✗ | ✔ |
| BEHAVIOR-1K(Li et al., 2023) | | ✗ | 1000 | ✗ | ✗ | ✗ | P | ✗ | ✗ | ✔ |
| EgoCOT(Mu et al., 2024) | | ✔ | 129 | ✗ | ✔ | ✔ | P | ✗ | ✗ | ✗ |
| EgoPlan-Bench(Chen et al., 2023) | | ✔ | 2406 | ✔ | ✔ | ✔ | P | ✗ | ✗ | ✔ |
| EgoPlan-IT(Chen et al., 2023) | | ✔ | 50K | ✔ | ✔ | ✔ | P | ✗ | ✗ | ✔ |
| HandMeThat(Wan et al., 2022) | | ✗ | 300K | ✔ | ✗ | ✔ | P/ G | ✗ | ✗ | ✔ |
| EgoVQA(Fan, 2019) | | ✔ | 520 | ✗ | ✗ | ✔ | P/ G | ✗ | ✗ | ✗ |
| EgoTaskQA(Jia et al., 2022) | EQA | ✔ | 40K | ✗ | ✗ | ✔ | P | ✔ | ✔ | ✗ |
| Egothink(Cheng et al., 2024) | | ✔ | 700 | ✗ | ✔ | ✔ | P | ✔ | ✔ | ✗ |
| OpenEQA(Majumdar et al., 2024) | | ✔ | 1600 | ✗ | ✔ | ✔ | P | ✗ | ✗ | ✔ |
| **ET-Plan-Bench** | TGP | ✔ | ∞ | ✔ | ✔ | ✔ | P/ FP/ G | ✔ | ✔ | ✔ |

(Valmeekam et al., 2024; Dziri et al., 2024). An automatic evaluation of LLM-based planners is crucial, yet performance comparison is complicated by factors such as prompt construction and model selection. Despite advances in task planning with LLMs, related benchmarks and automatic evaluations remain limited.

Evaluating LLM-based task planners requires datasets and simulators with task goal conditions, natural language instructions, and high-level APIs for simulator validation. ActivityProgram (Puig et al., 2018) provides household activity descriptions but lacks goal success conditions, relying on human annotators for success evaluation. Watch-And-Help (Puig et al., 2020), also based on Virtual Home (Puig et al., 2018), focuses on human-robot interactions but lacks natural language task descriptions. LoTa-BENCH (Choi et al., 2024) extends this with natural language in WAH-NL, yet the tasks remain simple, centered around object rearrangement with short action sequences. LLM-MCTS (Zhao et al., 2024) and EgoPlan-Bench (Chen et al., 2023) attempt to create complex tasks by combining simpler ones. LLM-MCTS uses the WAH dataset to define complex tasks just as combinations of simple tasks, while EgoPlan-Bench employs hierarchical reasoning, breaking tasks into subgoals. However, these approaches may not capture the real-world complexity of household tasks, as it fails to consider temporal or causal constraints between actions. Behaviour (Srivastava et al., 2022) sources diverse and complex household tasks from the American Time Use Survey, though its manual data generation limits scalability. HandMeThat (Wan et al., 2022) introduces ambiguity to tasks and includes an observational step, but interactions remain only to text. Conversely, ALFRED (Shridhar et al., 2020a) uses AI2-THOR (Kolve et al., 2017) and ALFWORLD (Shridhar et al., 2020b) to create visually complex, partially observable environments. RoboGen (Wang et al.,

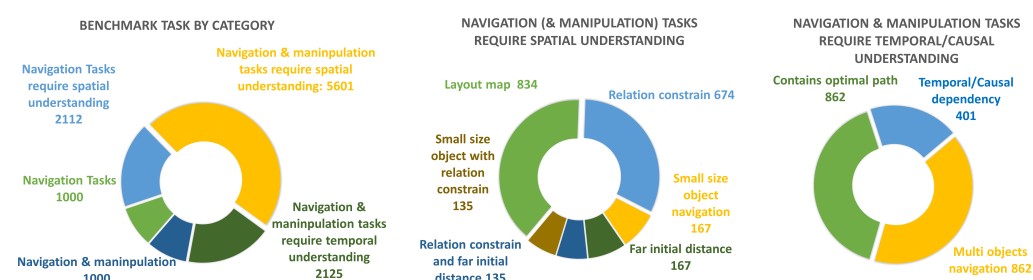

Figure 1: Evaluation task statistics of ET-Plan-Bench, which includes a diverse set of navigation and manipulation tasks, along with their advanced versions with spatial and temporal constraints.

2023) employs LLMs to generate infinite tasks and scenes for its simulation engine. Our work distinguished by its focus on generating complex tasks in a controlled manner, utilizing various types of constraints to create more realistic and challenging embodied tasks. To the best of our knowledge, no other work claims to offer a similar infinite task and data generation pipeline.

**Embodied question answering** Unlike end-to-end task planning and decomposition, embodied question answering tasks use egocentric visual signals to query environmental information (Fan, 2019; Majumdar et al., 2024) or formulate the task planning as a multiple question-answering problem (Chen et al., 2023; Mu et al., 2024). However, they do not allow interaction with the simulation environment for potential replanning when mistakes are made.

**Position of our work** We introduce a benchmark for household embodied task planning rather than question answering about the environment. This benchmark includes task descriptions and success criteria within a virtual environment. Any methods employing LLMs can be evaluated using our benchmark due to the substantial diversity in task descriptions, ensuring that LLMs will be employed for task decomposition at least once. Currently, the LLM component of the proposed LLM agent baseline focuses on skill-level task decomposition rather than lower-level control. Compared to other similar benchmarks on task planning, we design diverse tasks that impose various spatial, temporal, and causal constraints for navigation tasks and navigation & manipulation tasks. These tasks increase the level of complexity in a controlled manner and are able to diagnose foundation models' abilities in embodied task planning in a fine-grained fashion. While spatial and temporal constraints have been considered in the generation of embodied question-answering benchmarks such as OpenEQA (Majumdar et al., 2024), to the best of our knowledge, they have not yet been addressed in the context of embodied planning tasks. Additionally, we consider different levels of access to prior environment information, which determines the agent's knowledge about the state of the environment and is crucial for decision-making.

## 3 BENCHMARK DESCRIPTION

Since we propose a task generation pipeline, we can generate infinite task data given the preferred constraints and task difficulties. We include a subset of those task data as the evaluation task data. Figure 1 and Table 2 summarize our embodied planning evaluation benchmark, which includes navigation and manipulation tasks, along with their variants under spatial and temporal constraints.

### 3.1 BENCHMARK TASK DEFINITION

In our benchmark, we mainly provide three task categories: navigation tasks with or without spatial constraints, navigation and manipulation tasks with or without spatial constraints, and navigation and manipulation tasks with temporal constraints. For each task category, we divide it into several sub-task categories based on different aspects that require spatial and temporal understanding for the embodied agents. We introduce higher levels of task difficulty by combining the basic tasks. We further discuss the 5 key aspects that will impact the task difficulties in Appendix A.5.

**Navigation** Navigation tasks are common in everyday household scenarios, such as locating items. We define navigation tasks more specifically as object navigation, where the agent is required to navigate to an instance of a specified object category in unseen environments.

**Navigation with layout map** In real-world scenarios, robots often possess knowledge about certain environmental priors, such as a layout map which contains the locations of large pieces of furniture (e.g., sofas and refrigerators), which are less frequently moved compared to smaller objects. This information enables the robot to perform its navigation tasks more efficiently.

**Navigation with spatial relation constraint** This task requires the robot to identify a specific target object that meets given criteria, such as locating a box on the wall self. This spatial relation constraint could assess the LLM agent's perception ability of relative positions between objects. In general, these tasks are more challenging than navigation tasks without constraints.

**Navigation with occlusion due to small object size** 'Occlusion' refers to situations where large objects or barriers obstruct the view of target items, challenging an agent to provide strategies to uncover or reach these hidden target objects. This scenario is common in small object navigation, where the object will not be directly visible for the robot and complicating the robot's visual field and increase the difficulty of the tasks.

**Navigation with occlusion due to distant initial position** We designed tasks that target objects located far from the robot's initial position. The robot is then required to locate the target object over a greater distance. This setup poses greater challenges for the agent as it navigates through the room, with objects more likely to be obscured by large pieces of furniture.

**Navigation and manipulation** The most common embodied tasks, typically involve moving and then organizing items. For example, one might pick up one object and relocate it to a designated area, which could be a container or a furniture surface. Other task categories, such as navigation and manipulation with spatial constraints or occlusions, are similarly described in the earlier description.

**Navigation and manipulation with temporal and causal constraints** Compared to simply combining tasks that are order-invariant, temporal constraints involve actions that must be performed in a strict sequence. For example, an agent might need to prepare a meal where each step is executed in a specific order to ensure the dish is correctly completed. Alternatively, tasks may require the agent to optimize the order of actions to minimize the number of steps needed to complete a task. Additionally, some tasks feature Dependency Chains, where certain actions can only start once previous steps have been completed, such as needing to unlock and open a door before entering a room. Our designed navigation and manipulation tasks include both temporal and causal constraints. For example, the agent must sequentially place object A into container B, and then place object A and container B together into container C, in the correct order. Alternatively, the task requires the agent to sequentially find the target objects, grasp them, locate the recipient area, and place the object in the designated spot in the correct order. These tasks are crucial for assessing the agent's ability to understand the temporal and causal effects of environmental states and the actions it performs.

**Navigation and manipulation multiple objects in an optimal path with two arms** In navigation and manipulation tasks, a typical scenario involves preparing various items to achieve specific goals. This requires the robot to handle multiple objects simultaneously, such as gathering a pen and a notebook for writing. Virtual Home supports this by equipping the robot with two arms, allowing it to use either one or both arms to complete tasks. We design embodied tasks that specifically exploit these unique features. Using both arms can often be more efficient and optimal, for example, by reducing the moving distance and the number of exploration plan steps. We offer two versions for navigating and manipulating multiple objects: one-arm and two-arm configurations, enabling a comparison of their performance. We consider the two-arm version to be the optimal path.

## 3.2 TASK GENERATION USING LLMS

To efficiently generate a large and diverse set of tasks with various spatial and temporal constraints, we have incorporated large language models into the task generation pipeline. To enhance the generalizability of our task generation pipeline, each component can be adapted by a human to the specific simulator in use, incorporating a human-in-the-loop approach. Specifically, the pipeline includes the following steps, as illustrated in Figure 2:

Figure 2: Task generation pipeline. Task-specific requirements and scene graph from the simulator are used as inputs for automatic task generation. More details are given in Section 3.2

Table 2: Experiment results on our benchmark tasks

| Tasks | Success Rate | | Seq Length | | Longest Common Seq (Ratio) | | Moving Distance | |
|---|---|---|---|---|---|---|---|---|
| | GPT4 | LLAMA 7B SFT | GPT4 | LLAMA 7B SFT | GPT4 | LLAMA 7B + SFT | GPT4 | LLAMA 7B+ SFT |
| Navigation Tasks with or without Spatial Constraints | | | | | | | | |
| Navi + *Layout Map* | 90.77% | 91.13% | 3.76 | 3.96 | 1.32 (89.64%) | 1.34 (90.11%) | 10.79 | 10.83 |
| Navi | 79.26% | 80.58% | 6.77 | 6.75 | 1.59 (78.74%) | 1.62 (80.01%) | 14.10 | 14.59 |
| Navi + *Occlusion_Size* | 72.46% | 76.05% | 7.99 | 7.89 | 1.53 (74.95%) | 1.59 (78.14%) | 14.08 | 16.65 |
| Navi + *Occlusion_Distance* | 73.65% | 76.65% | 7.94 | 7.69 | 1.60 (77.40%) | 1.65 (80.24%) | 19.36 | 17.38 |
| Navi + *Relation* | 62.61% | 64.09% | 9.20 | 9.20 | 1.78 (88.75%) | 1.75 (86.05%) | 12.45 | 14.08 |
| Navi + *Relation + Occlusion_Size* | 60.74% | 61.48% | 9.68 | 9.91 | 1.74 (85.19%) | 1.70 (83.21%) | 13.26 | 16.05 |
| Navi + *Relation + Occlusion_Distance* | 54.81% | 55.56% | 10.41 | 10.31 | 1.73 (86.67%) | 1.67 (82.96%) | 15.75 | 16.22 |
| Navigation & Manipulation Tasks with or without Spatial Constraints | | | | | | | | |
| Navi & Mani + *Layout Map* | 83.98% | 83.96% | 12.36 | 12.09 | 4.20 (82.68%) | 4.17 (81.97%) | 22.22 | 21.67 |
| Navi & Mani | 73.76% | 74.33% | 17.02 | 16.47 | 4.17 (78.56%) | 4.22 (78.92%) | 28.51 | 26.99 |
| Navi & Mani + *Occlusion_Size* | 65.85% | 67.60% | 20.00 | 19.21 | 3.94 (75.00%) | 4.00 (75.27%) | 29.46 | 28.79 |
| Navi & Mani + *Occlusion_Distance* | 72.09% | 74.66% | 18.83 | 17.20 | 4.06 (75.50%) | 4.21 (78.08%) | 37.92 | 30.45 |
| Navi & Mani + *Relation* | 49.65% | 50.35% | 24.08 | 23.78 | 4.11 (73.60%) | 4.12 (72.14%) | 27.70 | 27.72 |
| Navi & Mani + *Relation + Occlusion_Size* | 43.03% | 42.75% | 26.52 | 26.39 | 3.81 (69.02%) | 3.82 (67.05%) | 31.55 | 28.42 |
| Navi & Mani + *Relation + Occlusion_Distance* | 49.88% | 50.00% | 23.96 | 23.79 | 4.20 (73.69%) | 4.25 (72.57%) | 34.85 | 30.83 |
| Navigation & Manipulation Tasks with Temporal Constraints | | | | | | | | |
| Navi & Mani + *Multi Objects* | 56.73% | 55.05% | 38.25 | 42.80 | 7.57 (69.92%) | 7.85 (69.09%) | 50.39 | 47.40 |
| Navi & Mani + *Multi Objects + Optimal Path with 2 Arms* | 72.04% | 74.21% | 28.12 | 28.51 | 5.60 (66.07%) | 5.61 (64.32%) | 38.25 | 39.17 |
| Navi & Mani + *Multi Objects + Temp Dependency* | 58.60% | 60.35% | 43.43 | 41.04 | 6.00 (64.05%) | 5.96 (62.71%) | 51.38 | 45.52 |

**Generating task template**. The first step is to define a task template for a specific type of task. Each template includes a task description and completion criteria. The description accounts for the number of objects involved, their properties, and any additional spatial or temporal constraints. This information is integrated as placeholders into various phrasings of the task. The completion criteria comprise the necessary state of relations between the elements involved at specific points during task execution. For example, for a task involving finding an apple and placing it in a fridge, the final success criteria would be (CLOSE, robot, fridge) & (INSIDE, apple, fridge).

**Gathering environment information from the simulator**. Given the specific task template, we gather related information about the simulation environments through its scene graph. We select all possible object candidates that satisfy the required constraint from the task template. For example, if the task template specifies finding objects with certain relational constraints such as "INSIDE", we will fetch the target objects and their corresponding objects that satisfy the "INSIDE" relation.

**LLMs assist task generation** After retrieving the relevant information from the environment, we shortlist all possible item combinations that meet the requirements of the commission template. For instance, the task of placing object A into container B necessitates retrieving both object A and container B from the environment. We subsequently employ LLMs to generate tasks adhere to principles of logical common sense and are rephrased in diverse ways to increase the diversity of

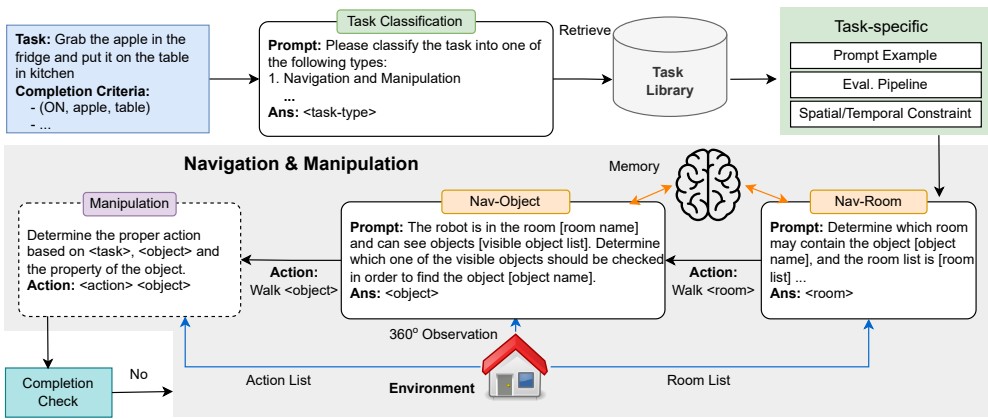

Figure 3: The LLM agent pipeline for evaluation comprises an automatic prompt selection module, a navigation module, and a manipulation module. More details are given in Section 4.1

descriptions. The LLMs receive both the environmental information and the task template as part of the prompt, which also includes rules and examples to ensure the tasks generated are realistic.

**Post-filtering based on the executability in simulator**. The tasks generated from the LLM are tested for their executability in the simulator. Any tasks that violate the physical rules encoded in the simulator are discarded and only executable tasks are kept.

**Ground truth planning label generation** We generate the ground truth planning sequence using the scene graph in each layout. We find the shortest path between the robot and the target items. A detailed elaboration of the ground truth action sequence generation is included in Appendix A.6.

Human evaluation is essential for ensuring the quality of generated tasks. To assess this, we randomly selected 5 simulation environments and chose 50 tasks from each environment. Human annotators then evaluated the reasonableness of these complex tasks. Our findings indicate that approximately 4% to 8% of the tasks were deemed unreasonable. The primary cause of the LLM's failure to correctly classify these tasks was ambiguity or misinterpretation in the descriptions of the objects involved. For instance, a task such as "Find the face cream tube and put it into the folder" might pass the LLM's post-checking process because the term "folder" could sometimes refer to a larger container. Overall, LLMs perform well in identifying unreasonable tasks.

# 4 BENCHMARK EVALUATION

## 4.1 OVERALL EVALUATION PIPELINE

While some existing methods rely on labor-intensive human evaluation (Huang et al., 2022), we propose an LLM agent baseline for automatic quantitative evaluation. As illustrated in the Figure 3, our LLM agent processes tasks through the following steps:

**Task classification** Firstly, the LLM agent determines the type of task, as listed in Section 3, based on the task description. Subsequently, tools and prompt examples specific to the identified task type are retrieved from the library and utilized in subsequent processes.

**Navigation-room** The object search algorithm employs a top-down hierarchical approach, beginning with the room where the target object is most likely to be found. The sequence of rooms to be searched is determined by the LLM agent, based on the task description and the list of rooms in the environment. For tasks involving spatial relation constraints, the room containing both the target object and the anchor object will be prioritized. If the anchor object is limited to a single room, only that room needs to be explored.

**Navigation-object** After arriving at a designated room, a 360-degree scan is conducted, and the objects within the field of view are identified by the perception module. If the target object is not

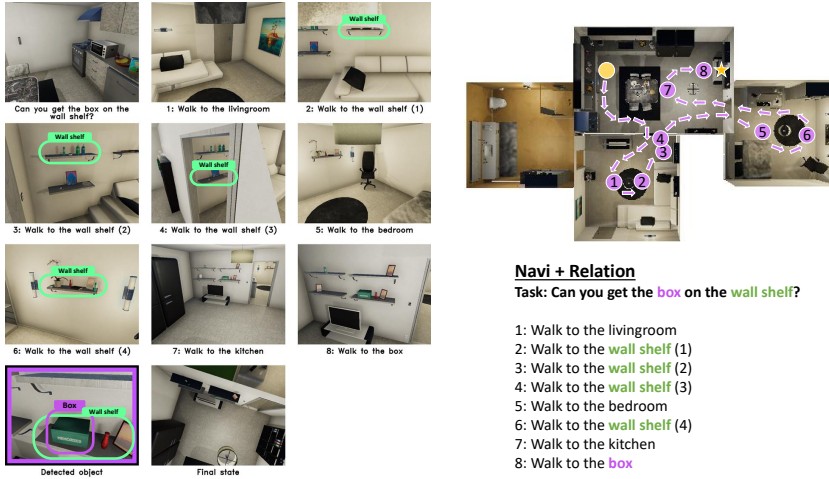

Figure 4: An example of a spatial constrained task. The robot explores different rooms and wall shelves to find one in which the relationship constraint matches the goal of the task. The images were generated using the Virtual Home simulator.

located, the LLM agent will actively select the most relevant visible object, such as large objects or pieces of furniture, for further exploration. The observed objects and their corresponding locations are stored in the agent's *memory*, and this information will be utilized to assist in locating subsequent objects if the task involves multiple items.

**Manipulation** For tasks involving the manipulation of target objects, the specific manipulation action will be executed as required by the LLM agent. Depending on the task requirements, the robot can employ one of several manipulation skills, including OPEN, CLOSE, GRAB, PUT, and PUTIN, to interact with the target object within the virtual environment.

**Completion check and iteration** The completion criteria will be evaluated after each navigation or manipulation step, as needed. If the criteria are not met, an additional round of navigation and manipulation will be carried out. The task will be deemed unsuccessful if the maximum number of action steps is reached before the task is completed.

## 4.2 EVALUATION RESULTS

Experiments were conducted in 8 NVIDIA Tesla V100 32G GPUs with Intel(R) Xeon(R) Gold 6140 CPU @ 2.30GHz. We use five evaluation metrics to evaluate the performance on our proposed embodied tasks, including success rate, action sequence length, longest common subsequence (LCS) length, LCS ratio with the ground truth action sequence, and moving distance for executing the tasks. A detailed explanation of these metrics is discussed in the Appendix A.9.

To verify the validity of our proposed benchmark and to demonstrate that tasks become more challenging after the addition of spatial or temporal constraints, we present comprehensive experimental results to explore the performance of our proposed LLM agent across various tasks in multiple simulators, including Virtual Home (Puig et al., 2018) and Habitat (Khanna et al., 2024). We analyze the performance on tasks of varying difficulty within the Virtual Home environment. In Appendix A.10, we examine whether Chain of Thought (CoT) prompting and few-shot in-context learning aid in task planning.

**Main results in Virtual Home** Table 2 displays the main results of various metrics for different types of tasks in Virtual Home simulator. In navigation tasks, the addition of spatial constraint dramatically impact the success rate. For occlusion cased by size or distance, we selected two subsets of tasks: The tasks of which the target object's size is the 20% smallest; And the tasks of which the distance between the robot's initial position and the target object position is top 20% largest. We observe a drop of 7% to 6% in success rate for both tasks, indicating that the occlusion caused by small object size or long distance increases the difficulty of the tasks. Consequently, the average sequence length

Table 3: Evaluation with open source and closed source LLMs including GPT-4, Llama3-70B, Mistral-8x7B and Llama3-8B.

| | Success Rate | | | |
|---|---|---|---|---|
| | GPT-4 | Llama3-70B | Mistral-8x7B | Llama3-8B |
| Navi + Layout Map | 90.77% | 89.04% | 89.04% | 83.99% |
| Navi | 79.26% | 77.39% | 75.84% | 69.66% |
| Navi + Relation | 62.61% | 66.06% | 62.41% | 27.74% |

and moving distance increase. Similar trend can be observed for tasks with spatial relation constraints. Combining relation and occlusion constraint is even more challenging for the agent to succeed.

The layout map significantly enhanced the success rate for both navigation and navigation & manipulation tasks. This improvement is attributed to the agent's prior global knowledge of large objects, such as substantial furniture, within the environment. Consequently, the agent could locate the target object more efficiently without necessitating extensive exploration.

It can be observed that when navigating and manipulating multiple objects, the success rate decreases, compared to that of one object. This indicates that as the sequence length of the task increases, there are more chances for the robot to make mistakes and fail the task. Besides, the average moving distance significantly decreases when the robot chooses the optimal path to manipulate multiple objects. The success rate also increases with the optimal path, suggesting that better planning is critical to the success of the tasks. When adding dependency to the task, the possible plans to complete the task get more limited, which enhances the difficulty level of the tasks. Therefore, the tasks with temporal dependency in average have lower success rate compared to the tasks completed with the optimal path.

**Results with additional LLMs** To assess the impact of various LLMs on the evaluation process, we have additionally evaluated the benchmark with more LLM models as shown in the Table 3. While other powerful open-source LLMs like Llama3-70B and Mixtral-8x7B show similar performance as GPT-4, smaller model Llama3-8B struggles with navigation task, with only 27.74% success rate for navigation tasks with spatial constraint.

**Supervised finetuning** To evaluate whether generated data can enhance the ability of smaller LLMs to do embodied reasoning and follow instructions better, we introduce additional data from other layout environments to conduct supervised fine-tuning (Tasks from 34 room layout for training, 5 for validation and 10 for evaluation, please refer to Appendix A.7 for details). This setup allows us to assess generalization across different environments. Using the data generation pipeline described in Section 4.1, with the assistance of GPT-4 (Achiam et al., 2023), we generated question-and-answer (QA) pairs and executable action plans from the training data. These QA pairs were then used for supervised fine-tuning (SFT) a smaller open-source LLM, LLAMA2-7B (Touvron et al., 2023), and its performance was compared with other LLMs, such as GPT-4. Furthermore, this generated data has the potential to enhance foundation models' capabilities in both real-world understanding and task decomposition.

The performance of SFT LLAMA closely matches that of GPT-4. In most cases, SFT LLAMA slightly outperforms GPT-4, primarily because SFT LLAMA tends to identify at least one large object that allows the robot to explore, whereas GPT-4 may indicate that no visible objects need exploration in these tasks. The training data teaches the LLMs to actively explore the environment. This results in SFT LLAMA having a longer moving distance than GPT-4 in navigation tasks, thereby increasing the likelihood of finding the target object through more comprehensive exploration.

**Results in Habitat** In addition to the Virtual Home environment, we conducted experiments on navigation tasks using another widely used household task simulator, Habitat 2.0 (Figure 5). The results presented in Table 4 demonstrate that our fine-tuning strategy significantly enhances the performance of small-scale LLMs, enabling them to achieve a level comparable to state-of-the-art models like GPT-4. Moreover, the fine-tuned LLMs performs especially well on instruction following.

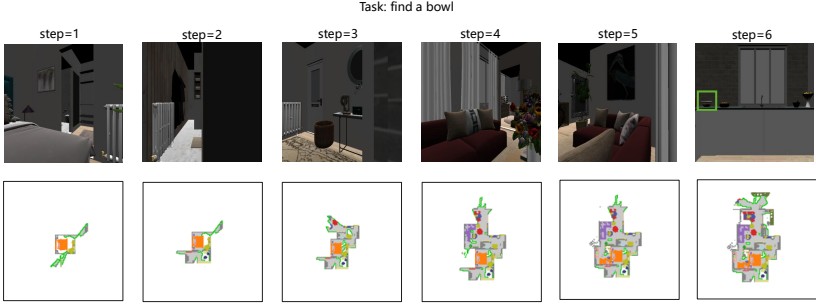

Figure 5: In a mapless task scenario, the robot begins in the bedroom, gradually explores the living room and kitchen, and ultimately discovers the bowl. The images were generated using the Habitat 2.0 simulator.

Table 4: Comparison for results from different simulators.

|  | Navigation Tasks (Virtual Home 3.0) | Navigation Tasks (Habitat 2.0) |
|---|---|---|
| GPT-4 | 79.26% | 72.2% |
| LLAMA - SFT | 80.58% | 71.9% |

## 4.3 CASE STUDY

Figure 4 illustrates one example of successful cases where the robot explores different rooms before finding the target box with relation constraint. Some failed cases are shown in Appendix A.2.

Some instances of failure are attributable to defects in the perception module, which is provided by the Virtual Home API. In addition, the target object could not be found even if it is very close, if the robot is not directly facing it. The spatial relation constraints poses extra challenges. For example, if the task requires finding a glass near a monitor, the monitor might block the glass from the robot's view, preventing it from correctly verifying the spatial relationship. Detailed explanation can be found in Appendix A.2.

## 5 CONCLUSION

Our study has some limitations worth noting. The evaluation was conducted in two virtual environments. While we can benefit from controlled experimentation, it might not fully capture the complexities of real-world settings, potentially leading to a sim-to-real gap. Furthermore, there are areas within the perception module that could benefit from further exploration and improvement.

In this work, we present an automatic embodied planning task generation and an LLM agent baseline for benchmark evaluation. We introduce spatial and temporal constraints in navigation and manipulation tasks, which are barely touched by existing embodied planning benchmarks. The method is scalable and can generate infinite number of diverse embodied planning data. The experiment results demonstrate that addition of spatial and temporal constraints, which are common in real world embodied tasks, poses significant challenge to the agent with the most advanced LLM GPT-4, because the success rate dramatically reduces for the tasks involving spatial or temporal constraints. Through supervised finetuning, the much smaller model like LLAMA-7B can match or even slightly outperform GPT-4. Given that we merely introduced an LLM agent as a baseline for benchmark evaluation, there remains considerable potential for baseline improvement in future research. Overall, our proposed benchmark can assist researchers in evaluating the performance of their LLM agents in completing complex embodied planning tasks that involve spatial and temporal constraints.

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

# A APPENDIX

## A.1 SUCCESSFUL CASES OF EMBODIED PLANNING TASKS

Some successful cases of different types of planning tasks are shown in figs. 6 to 15. We added extra annotations as task type and bounding boxes. All of our examples highlight the main object(s) in both the task description and figure with different colors for a clearer understanding on what the goal is and which actions followed up to complete it. Not all of our examples have relation constrains, but for those that have, we highlight those relations with different bounding boxes and dotted lines to distinguish different objects in the scene. As well, for those examples whose relation is set as "object A facing object B", we also provide two perspective views to frame the spatial relation between the objects.

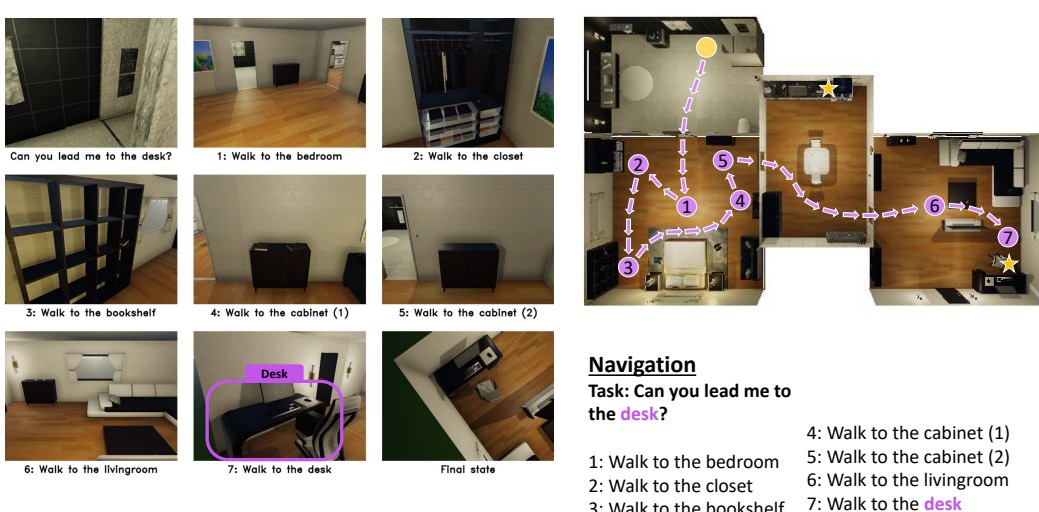

Figure 6: An example of a simple navigation task. The robot explores the rooms and successfully finds the desk.

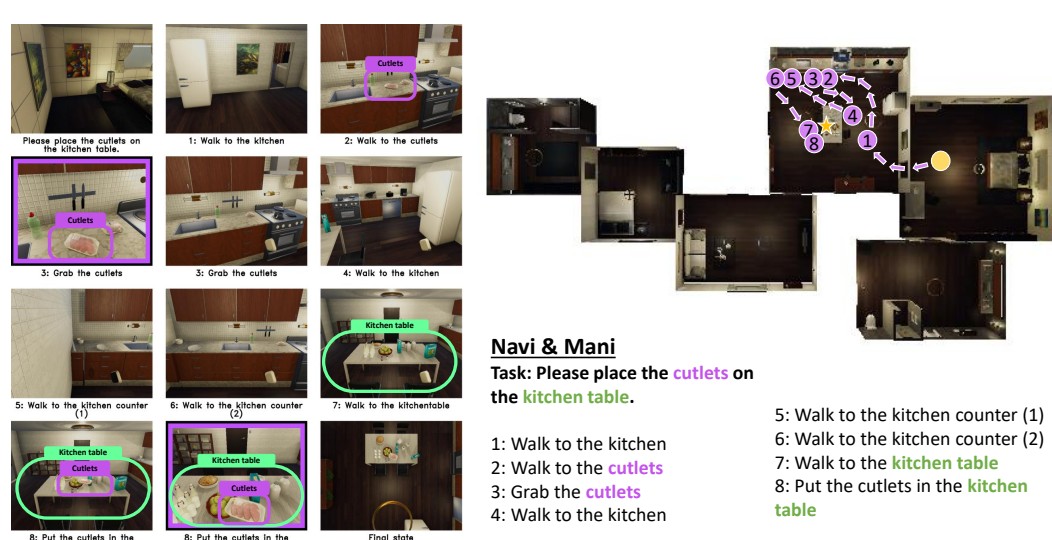

Figure 7: An example of a simple navigation and manipulation task. The robot successfully finds the cutlets and then places them on the kitchen table.

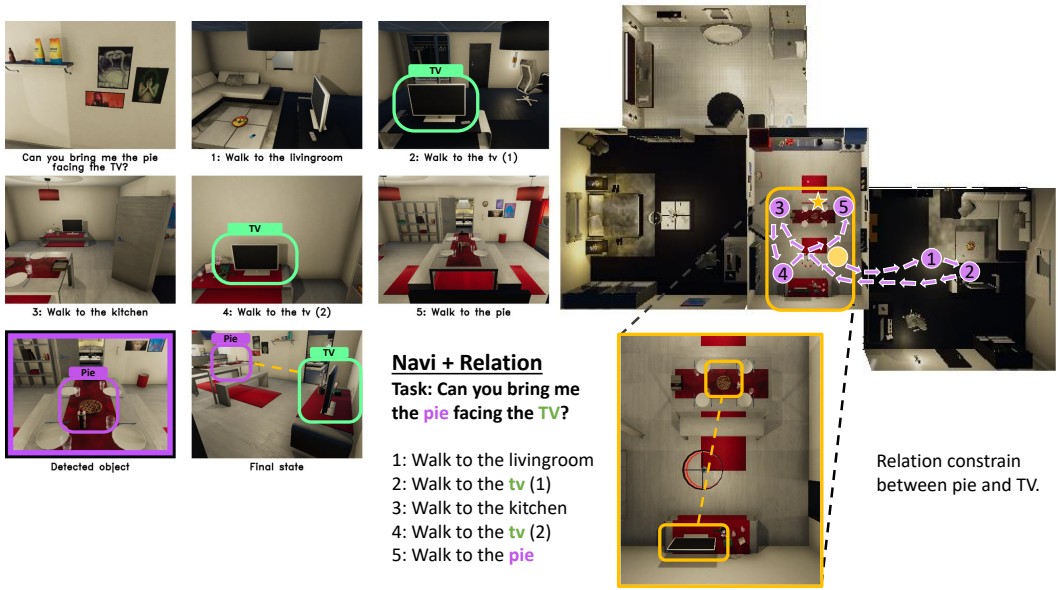

Figure 8: An example of a navigation task with spatial relation constraint. The robot successfully finds the pie facing the TV.

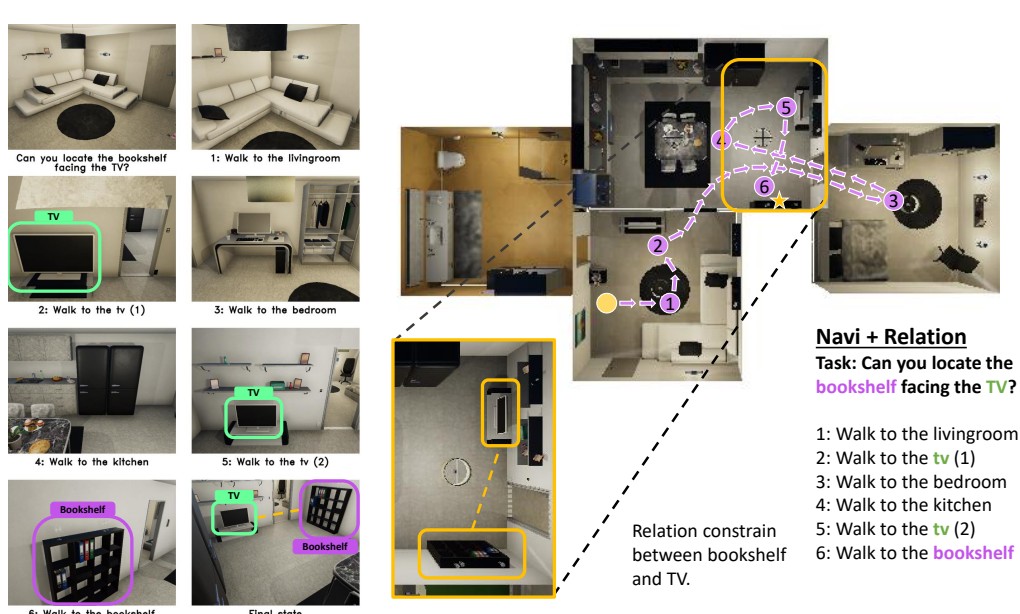

Figure 9: An example of a navigation task with spatial relation constraint. The robot successfully finds the book shelf facing the TV.

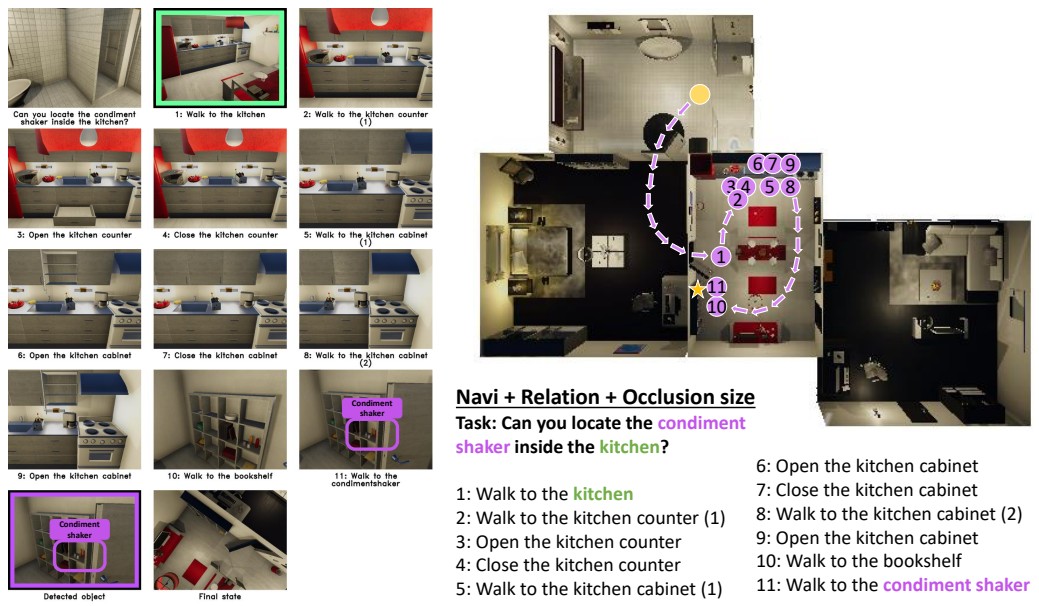

Figure 10: An example of a navigation task with spatial relation and size constraint. The robot tries to locate a small object inside of the cabinets, and find it among other small objects on the book shelf.

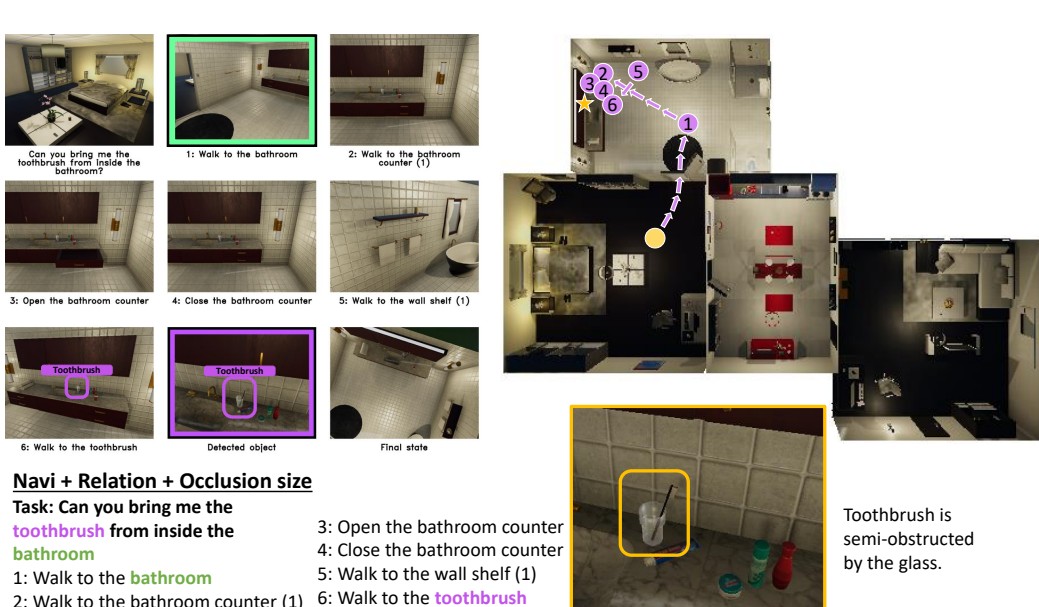

**Navi + Relation + Occlusion size**

**Task: Can you bring me the toothbrush from inside the bathroom**

1: Walk to the **bathroom**
2: Walk to the bathroom counter (1)
3: Open the bathroom counter
4: Close the bathroom counter
5: Walk to the wall shelf (1)
6: Walk to the **toothbrush**

Toothbrush is semi-obstructed by the glass.

Figure 11: An example of a navigation task with spatial relation and size constraint. The robot tries to locate a toothbrush, and find it finally in the bathroom counter, partially obscured by the glass. The robot has to walk to the counter before the toothbrush can be seen.

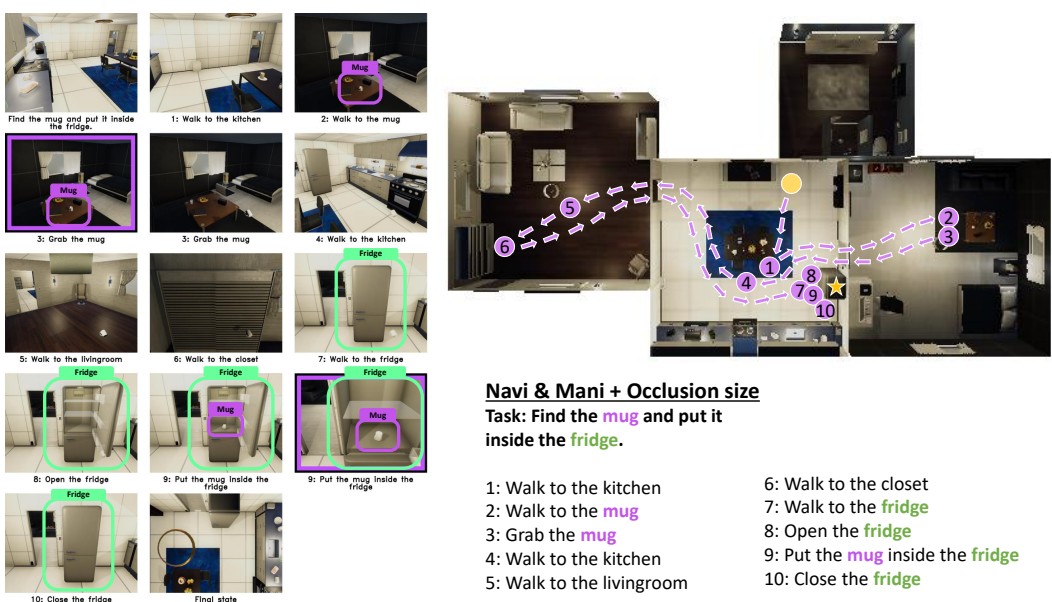

**Navi & Mani + Occlusion size**

**Task: Find the mug and put it inside the fridge.**

1: Walk to the kitchen
2: Walk to the **mug**
3: Grab the **mug**
4: Walk to the kitchen
5: Walk to the livingroom
6: Walk to the closet
7: Walk to the **fridge**
8: Open the **fridge**
9: Put the **mug** inside the **fridge**
10: Close the **fridge**

Figure 12: An example of a navigation and manipulation task with size constraint. The robot locates a small object, a mug, and place it insider a container, the fridge.

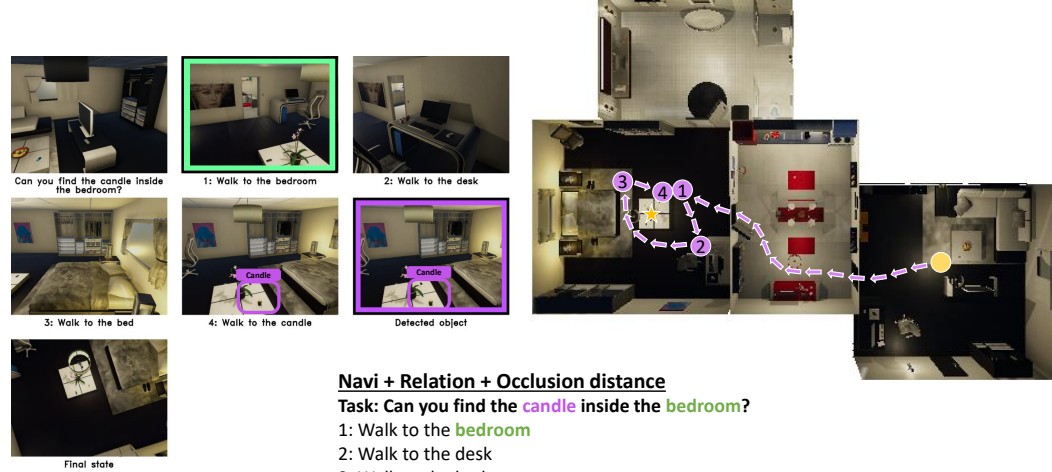

**Navi + Relation + Occlusion distance**
Task: Can you find the candle inside the bedroom?
1: Walk to the bedroom
2: Walk to the desk
3: Walk to the bed
4: Walk to the candle

Figure 13: An example of a navigation task with distance constraint. The robot has to walk from the living room to the bedroom on the other side of the apartment to find the candle placed in a table inside the bedroom.

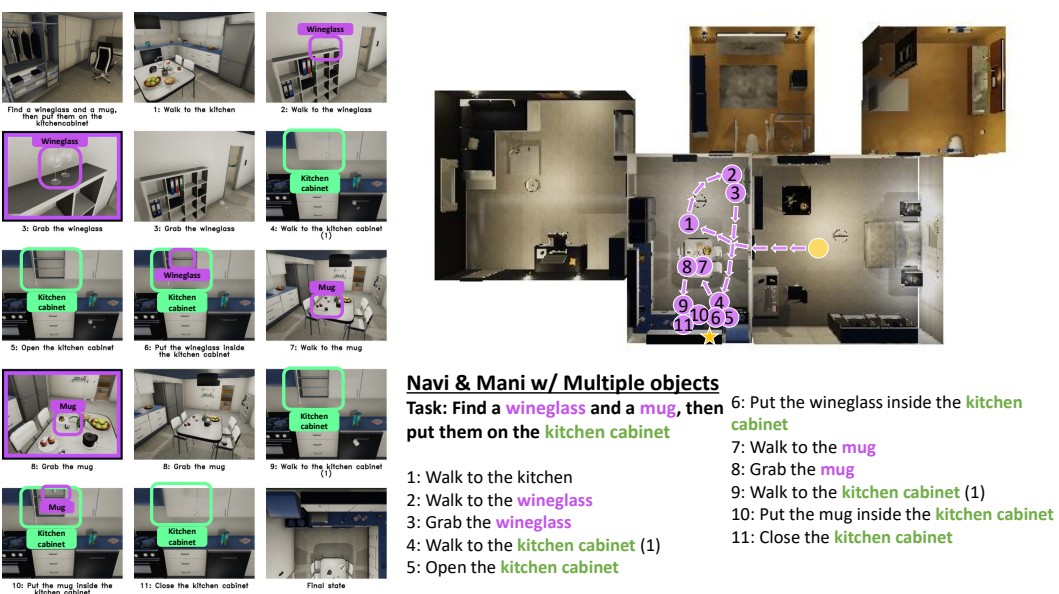

**Navi & Mani w/ Multiple objects**
Task: Find a wineglass and a mug, then put them on the kitchen cabinet

1: Walk to the kitchen
2: Walk to the wineglass
3: Grab the wineglass
4: Walk to the kitchen cabinet (1)
5: Open the kitchen cabinet

6: Put the wineglass inside the kitchen cabinet
7: Walk to the mug
8: Grab the mug
9: Walk to the kitchen cabinet (1)
10: Put the mug inside the kitchen cabinet
11: Close the kitchen cabinet

Figure 14: An example of a navigation and manipulation task involving multiple objects. The robot has to find both objects and put them on the kitchen cabinet. The robot completes this task using only one arm.

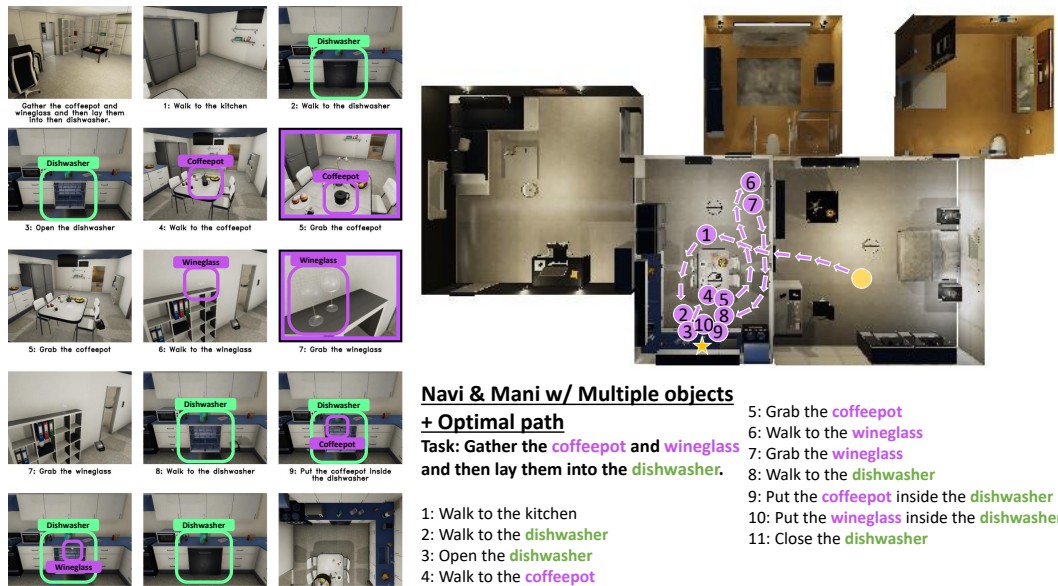

Figure 15: A navigation and manipulation task involving multiple objects. In this scenario, the robot successfully completes the task in an optimal path. It firsts opens the dishwasher where the objects needs to be placed, then grab both objects using both arms, and finally place them inside the dishwasher.

## A.2 FAILED CASES OF EMBODIED PLANNING TASKS

Some failed cases of the planning tasks are shown in figs. 16 to 19. For these instances of failure, a cropped view and a brief description were added to elucidate the reasons behind the task's failure.

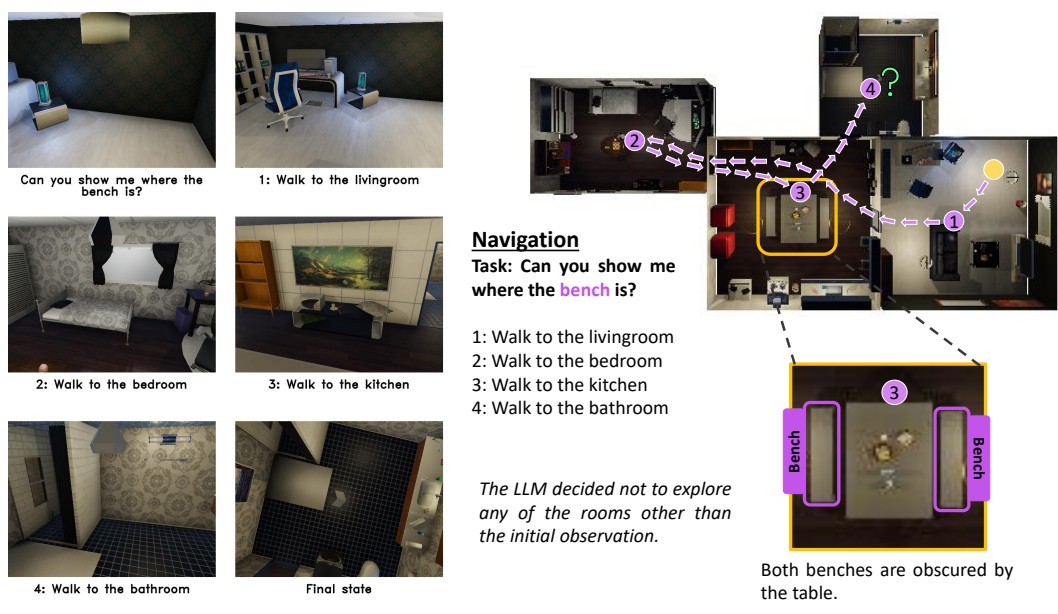

Figure 16: An example of a failed task caused by occlusion. Both benches are obscured by the table and cannot be seen by the robot in this case.

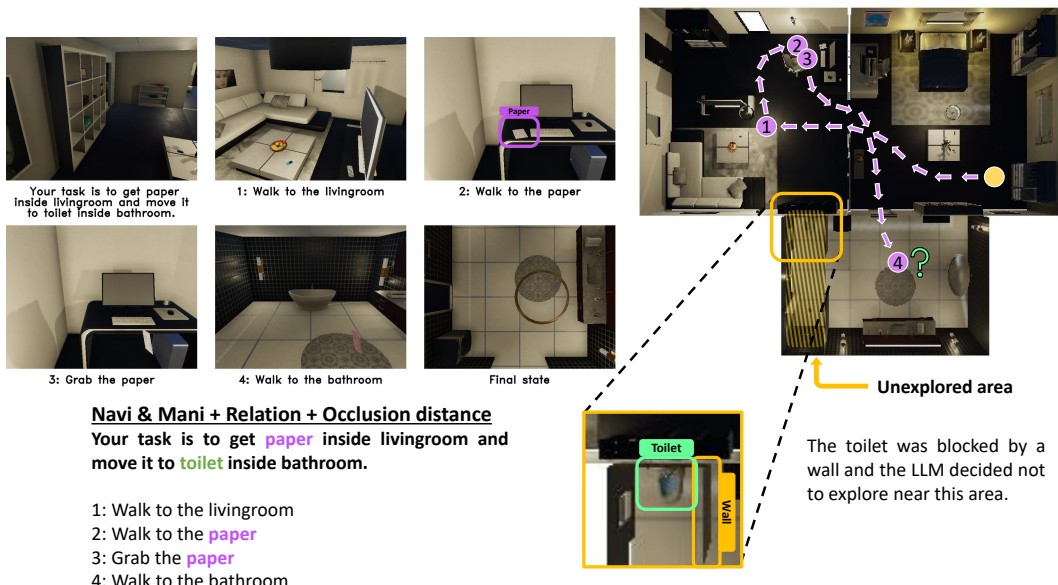

Figure 17: An example of a failed task caused by occlusion. The toilet is obstructed by a wall.

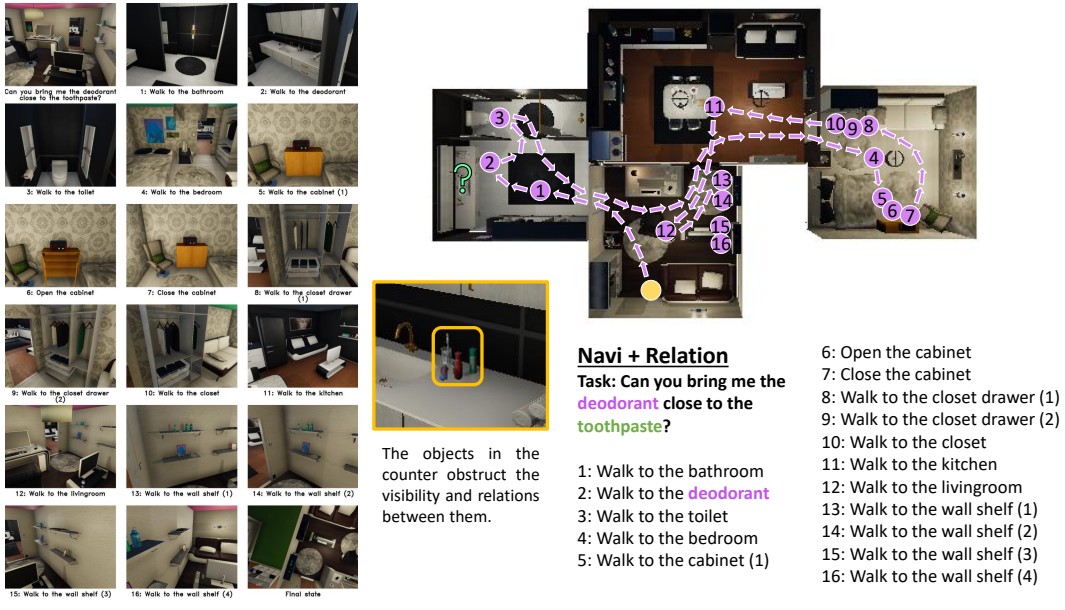

Figure 18: An example of a failed task due to occlusion is when the robot detects the deodorant but is unable to see the toothpaste, as it is obstructed from view.

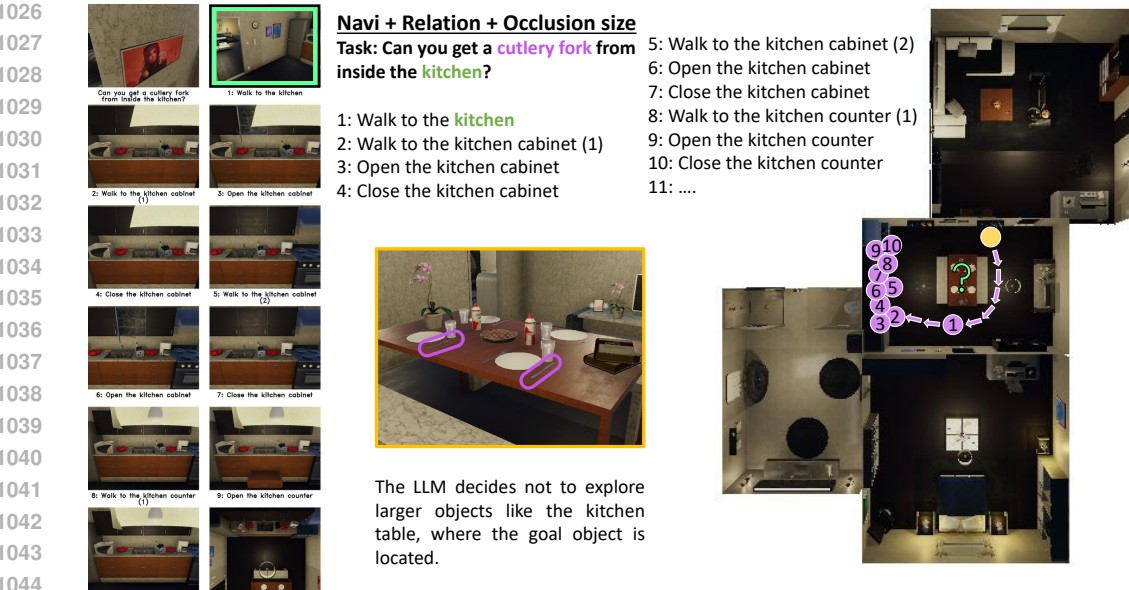

Figure 19: An example of a failed task due to inefficient exploration involves a robot tasked with finding a cutlery fork in a kitchen. Despite the fork being located on the kitchen table, the robot only explores various containers that might contain the cutlery. As a result, it exhausts its maximum exploration steps before it has a chance to search the table.

### A.3 A COMPREHENSIVE RELATED WORK SURVEY

#### A.3.1 EVALUATION OF FOUNDATION MODELS FOR HOUSEHOLD EMBODIED PLANNING

LLMs are used in task planning for their generalization capabilities across various tasks(Brown et al., 2020; Ahn et al., 2022; Huang et al., 2022). Recent works show significant advances in using LLMs for long-horizon planning (Zhao et al., 2024), but challenges like hallucination and poor spatial reasoning remain (Valmeekam et al., 2024; Dziri et al., 2024). An automatic evaluation of LLM-based planners is crucial, yet performance comparison is complicated by factors such as prompt construction and model selection. Despite advances in task planning with LLMs, related benchmarks and automatic evaluations remain limited.

Evaluating LLM-based task planners requires datasets and simulators that include goal conditions of tasks, natural language instructions, and a high-level API that facilitates validation using the simulator. ActivityProgram (Puig et al., 2018) is a dataset of household activity descriptions in Virtual Home (Puig et al., 2018). However, it does not include goal conditions, leaving the evaluation of task success to human annotators. Watch-And-Help (Puig et al., 2020) (WAH), another dataset based on Virtual Home (Puig et al., 2018) focusing on human-robot interactions, lacks natural language descriptions of the household tasks. LoTa-BENCH (Choi et al., 2024) proposes WAH-NL, an extension to the Watch-And-Help dataset that includes natural language descriptions. However, all tasks in both datasets are generally about object rearrangement and only have a similar level of complexity with short sequences of actions.

LLM-MCTS (Zhao et al., 2024) attempts to create more complex tasks using the WAH dataset by defining a complex task as a composition of multiple simple tasks. However, combining simple tasks does not necessarily increase the difficulty of planning, and this approach does not represent the complexity of household tasks in real-world cases. EgoPlan-Bench (Chen et al., 2023) also provides hierarchical reasoning for tasks, claiming that the goal can be broken down into subgoals and secondary subgoals. However, it still follows the idea that combining simpler tasks creates more complex tasks, ignoring the potential constraints and obstacles that can make tasks more challenging in a real-world setting.

Instead of generating tasks, Behaviour (Srivastava et al., 2022) takes household activities from the American Time Use Survey, focusing on the diversity and complexity of tasks. However, the generation of data is not automatic, rendering it hard to scale. HandMeThat (Wan et al., 2022) proposes another benchmark dataset to add ambiguity to the generated tasks, forcing the agent to retrieve more implicit information from the task. Like WAH (Puig et al., 2020), HandMeThat (Wan et al., 2022) also includes a watch step where the agent observes actions performed by humans. They generate their tasks by instantiating them from the human-annotated tasks in Behaviour (Srivastava et al., 2022). However, they only interact with a text-based environment.

Datasets such as ALFRED (Shridhar et al., 2020a) are also generated to benchmark embodied planning tasks using AI2-THOR (Kolve et al., 2017) or ALFWORLD (Shridhar et al., 2020b) by generating more visually complex scenes of partially observable environments. RoboGen (Wang et al., 2023) adopts a generative approach to generate diverse tasks, simulation scenes, and training supervisions. They leverage Large Language Models for generating tasks, scene configurations, and task decomposition and finally deploy the generated scenes in a generative simulation engine.

In this research, we focus on household activity planning, which is a combination of object navigation and manipulation. There is other research more focused on lower-level manipulation tasks that we did not mention here, for example ManiSkill2 (Gu et al., 2023), which is out of the scope of high level task planning.

### A.3.2    EMBODIED QUESTION ANSWERING

Ego-centric Visual Question Answering (EgoVQA) is a crucial task for many robotic applications. EgoVQA (Fan, 2019) is a benchmark dataset focusing on first-person VideoQA tasks instead of third-person videos. They collected and manually annotated the egocentric VideoQA pairs. EgoTaskQA (Jia et al., 2022) is a dataset of question-answer pairs for ego-centric videos of human activities; however, it specifically focuses on descriptive, predictive, explanatory, and counterfactual questions, attempting to understand spatial, temporal, and causal relationships in the tasks. OpenEQA (Majumdar et al., 2024) is another benchmark that includes human-generated questions about the environment with open-vocabulary answers. The questions are categorized into two main types: those where the agent needs to use episodic memory, such as video, to extract the answer, and those where the agent needs to explore the environment to find the answer. They leverage LLMs as evaluators to score the answers.

The task of Embodied Question Answering (EQA) has also influenced task planning. EgoPlan-Bench (Chen et al., 2023) designs the planning task as a multiple question-answering problem, and EgoCOT (Mu et al., 2024) constructs the task planning data by generating questions about how the task must be planned. In this work, we focus on task planning using different levels of complexity rather than answering questions about the environment. However, we continue to employ a question-and-answer format to prompt the Large Language Model to generate a plan for the task.

### A.4    DETAILS FOR EACH TASK GENERATION PROCESS AND BENCHMARK TASKS

We provide several categories of concrete tasks in our benchmark: navigation; navigation with spatial constraints; navigation and manipulation; navigation and manipulation with spatial constraints; and navigation and manipulation with temporal constraints. The generation of each task follows the pipeline described in Section 3.2. The completion criteria, description and examples used for each task are displayed in Table 5. The details of each task description are provided as following:

**Navigation**. This category of tasks consists of easy tasks in which the robot needs to find a certain object. The LLM agent determines where the object might appear and guides the robot to explore the room until the task is successfully completed.

**Navigation + Relation**. In this category of tasks, the robot is asked to find a certain object in a specific place. The LLM agent needs to provide answers about where the object might be, considering the location where the object is situated.

**Navigation & Manipulation**. This type of task requires the robot to find a specific object, grab it, and place it inside a container or on the surface of another object or the piece of furniture. The large language model needs to help the robot locate both objects. Since there are more possibilities for the stalling of the robot and it is harder to find the objects, the LLM agent is required to solve a more complex reasoning problem.

Table 5: Task description.

| Task Category | Description | Example | Intermediate Criteria | Success Criteria |
|---|---|---|---|---|
| *Navigation* | Find an object | Find an apple. | - | (CLOSE, robot, apple) |
| *Navigation + Relation* | Find an object at certain place | Find the apple in a fridge. | (INSIDE, apple, fridge) | (CLOSE, robot, apple) |
| *Navigation & Manipulation* | Find an object and put it inside or on an another object | Find an apple and put it in a fridge. | - | (CLOSE, robot, apple)(CLOSE, robot, fridge)(INSIDE, apple, fridge) |
| *Navigation & Manipulation + Relation* | Find an object at certain place and put it in or on another object at certain place. | Find an apple close to a cup and put it in a fridge in a kitchen. | (CLOSE, robot, apple)(CLOSE, apple, cup) | (CLOSE, robot, fridge)(INSIDE, fridge, kitchen)(INSIDE, apple, fridge) |
| *Navigation & Manipulation + Multiple objects* | Find multiple objects and put them inside or on another object | Please obtain toothpaste and toothbrush and place on the kitchen counter | - | (CLOSE, robot, toothpaste) (CLOSE, robot, toothbrush)(CLOSE, robot, kitchen-counter)(ON, toothpaste, kitchen-counter)(ON, toothbrush, kitchen-counter) |
| *Navigation & Manipulation + Temporal Dependency* | Find and place multiple objects with logical dependency | Find milk and pour it into a cup and put it into the oven to heat it up. | (CLOSE, robot, milk)(CLOSE, robot, cup)(INSIDE, milk, cup) | (CLOSE, robot, microwave)(INSIDE, cup, microwave)(INSIDE, milk, cup) |

**Navigation & Manipulation + Relation**. In this type of task, the robot is asked to find object A, which is related to object B in the environment, grab it, and then place it inside or on object C, which is related to object D in the environment. This relational constrain increases the difficulty of finding the objects to complete the task. Therefore, the LLM agent must consider the relationship between objects to make decisions about the robot's plan.

**Navigation & Manipulation + Multiple objects**. Rather than locating multiple objects (two or more), grabbing each one sequentially (each grab is an independent sub-task), and then placing them on or inside another object, the LLM agent will tackle the challenge of optimizing the traversal path and the order of completing each subtask, aiming to complete the tasks more efficiently.

**Navigation & Manipulation + Temporal Dependency**. Each task consists of a sequence of subtasks, where each subtask must be completed before the next one can begin. This results in a more strict planning path for the LLM agent. Therefore, the large language model needs to think more logically to guide the robot in exploring the environment and completing the task. The prompts used for LLM to filter out unreasonable tasks are illustrated in Table 8 and Table 9.

A.5    Factors that Impact Embodied Task Difficulties

Based on tasks described in Table 2 of the main paper, we categorize them into various difficulty levels according to 5 aspects: action sequence length, prior knowledge, spatial relation constraints, occlusion, and temporal constraints.

**Action sequence length**    Based on the action sequence length, we categorize our benchmark dataset into three sub-datasets: 1. Navigation tasks with or without spatial constraints: These are the simplest tasks as they only require the agent to find the target object. Reaching the target object requires at least one step. 2. Navigation and manipulation tasks with or without constraints: These tasks are more challenging as they necessitate not only finding the target object and the recipient location, but also moving the target object to the recipient location. Completing the task requires at least four steps: walking to the target object, grabbing the target object, walking to the recipient location, and placing the target object at the recipient location. 3. Navigation and manipulation multiple objects: These are the most complex tasks, requiring the agent to locate multiple target objects and the recipient location, move all objects to the recipient location sequentially, and determine whether the recipient location can accommodate both objects simultaneously.

**Prior knowledge** Prior knowledge is categorized into three levels of environmental semantic information: visible information, visible information supplemented with a layout map, and comprehensive environmental information with full scene graph. "Visible information" implies that the agent can only obtain details about the environment through its embodied vision capabilities, simulating a real-world scenario in which a robot explores an entirely unknown environment from scratch. This scenario presents the most challenging task for the robot to complete. In another real-world scenario, the robot might possess knowledge of certain environmental details, such as the locations of large, immovable furniture. This additional information makes it relatively easier for the robot to complete the task. In the final scenario, the robot has detailed information about the entire environment including the location of all small objects, making it the easiest task to complete, as the robot can navigate directly to its destination without the need for any exploration.

**Spatial relation constraints** Tasks with spatial relation constraints are more challenging than those without because verifying the spatial relationship between two objects requires advanced scene understanding abilities of the agent. However, in some scenario, these constraints also provide additional information, facilitating more efficient exploration.

**Occlusion** As discussed in section 3.1, tasks involving occlusion are more challenging, either because the size of the object is small or because the initial distance between the robot and the target object is far, compared to those without occlusion.

**Temporal constraints** Tasks involving the navigation and manipulation of multiple objects with temporal constraints are more challenging than those without, as the robot must adhere to a specific sequence of steps. Unlike tasks with completion criteria only at the final stage, these tasks include various intermediate completion criteria due to their causal dependencies.

## A.6    Ground Truth Generation

For each task, we are able to generate ground truth data directly from the scene graph of the layout provided in the simulator, where objects are represented as nodes and the relationships between them are represented as edges. For example, in order to find an apple located inside a fridge in the kitchen, the shortest path between the robot and the apple can be extracted from the scene graph: kitchen-fridge-apple. Then the path is translated into sequence of actions which could be executed in the simulator. If the action list can be successfully executed and the goal conditions are met, the data is saved as ground truth that represents the shortest action list to complete the task. The tasks that cannot be completed in this way are removed from the dataset. This happens mostly due to inherent bugs of the simulator.

The environment may contain multiple instances of objects with the same name. For tasks involving spatial constraints, the target object meeting the specified requirements is identified. Utilizing the unique object ID, the shortest path to the specific target object is then determined. For the tasks that require object manipulation, the actions of "grab" or "place" are added to the target object or the recipient location after navigation.

The tasks with temporal constraints, such as firstly placing object A inside object B then placing object B that contains A into object C, can be decomposed into two sub-tasks: (1) find object A and put it in object B; (2) grasp object B and put it into object C. The ground truth for each sub-task can be generated using the aforementioned logics.

For tasks involving navigation and manipulation of multiple objects (e.g., placing both object A and object B at location C), the single-arm approach generates the ground truth by moving the two target objects to the destination one by one. It is a straightforward combination of two navigation & manipulation tasks involving a single object. In contrast, the two-arm approach involves first locating both objects, then proceeding to the destination while carrying one object in each hand, and finally placing them down.

## A.7 TRAINING, VALIDATION AND TESTING DATA FOR OUR SFT EXPERIMENTS

In order to evaluate the generalization ability across different environments, We partitioned tasks based on Virtual Home environments. We use tasks from 34 environment for training, 5 for validation, and 10 for testing and evaluation. We used the LLM agent baseline on the training tasks to generate question-and-answer (QA) pairs and executable action plans (see one example details in Table 10). To test the effectiveness of the generated data, we fine-tuned LLAMA-7B using these QA pairs, and compared its performance with GPT-4.

To evaluate the generalization ability across various task types, we employed QA data, comprising a total of 5457 QA pairs, from four types of training tasks: navigation, navigation with spatial relation constraints, navigation & manipulation, and navigation & manipulation with spatial relation constraints. This data was utilized to fine-tune the LLAMA-7B model. Subsequently, we integrated the fine-tuned LLAMA-7B model into the LLM agent to evaluate all testing tasks.

The validation data is used for hyper-parameter tuning for SFT. Results for the testing tasks are presented in Table 2 of the main paper. In future works, these data could potentially used to enhance the foundation models' capabilities in long-horizon task decomposition in real-world environment.

## A.8 EVALUATION DETAILS FOR EACH TASK

Our benchmark provides the completion criteria for each task. We developed an LLM agent to interpret embodied planning tasks with various constraints and effectively follow instructions. The initial phase involves task classification, which is accomplished using the prompt displayed in the Table 11.

The LLM agent's exploration strategy involves locating the target object using its vision, navigation, and manipulation capabilities. Virtual Home offers an API to access the visible graph, which details the objects within the field of view and their relationships from a first-person perspective. It also enables navigation through the execution of the WALK action, specified by the object name and ID. Additionally, the robot has various manipulation skills such as OPEN, CLOSE, and GRAB during its exploration. The core algorithm governing the robot's exploration logic is detailed in Algorithm 1. The primary idea involves systematically exploring rooms and potential large visible objects one by one in a 360-degree manner until either the target object is found or the maximum exploration steps are reached. The prompt template for ranking rooms is presented in Table 12, while the prompt template for the selection of large objects is shown in Table 13. In Virtual Home, each TURNRIGHT action corresponds to a 30-degree right turn. Therefore, we perform the TURNRIGHT action 12 times to simulate a complete 360-degree exploration. The LLM is involved in ranking rooms and the selection of large objects, thereby enhancing the agent's cognitive capabilities.

In tasks involving navigation with spatial constraints, the main concepts are similar to those in Algorithm 1, with two key differences: 1. After locating the target object, the robot must verify if the object meets the spatial constraint criteria. 2. If the robot observes only the target object or the constrained object, it should proceed to investigate one of these objects further, instead of exploring other large pieces of furniture. Detailed steps of this algorithm are provided in Algorithm 2.

The LLM agent for evaluating navigation and manipulation tasks is detailed in Algorithm 3. This task can be broken down into four subtasks: navigating to the object to be manipulated, grasping the object, navigating to the recipient location, and placing the object at the recipient location. The grasping and placing actions can be executed using the Virtual Home GRAB, PUT, or PUTIN actions. The navigation to the object to be manipulated follows the procedure outlined in Algorithm 1. During the navigation process, all visible objects are stored in "memory". If the recipient location is already in memory, the robot can navigate there directly, avoiding redundant exploration. Otherwise, the robot

must locate the recipient by following Algorithm 1 again. Additionally, the LLM agent for evaluating navigation and manipulation with spatial constraints combines Algorithm 2 and Algorithm 3.

The aforementioned tasks involve the navigation and manipulation of a single object. For the tasks that involve navigating and manipulating two objects, such as placing object A and object B at recipient location C, the task can be decomposed into the following eight sequential steps: 1. Navigate to object A, 2. Grab object A, 3. Navigate to recipient location C, 4. Place object A at recipient location C, 5. Navigate to object B, 6. Grab object B, 7. Navigate to recipient location C, and 8. Place object B at recipient location C. The navigation steps can follow Algorithm 1, while Virtual Home provides the necessary manipulation skills for the task.

Given that the character in Virtual Home can manipulate objects using both arms, we propose a task decomposition pipeline that entails navigation and the use of two arms to handle two objects simultaneously. This process differs slightly from single-arm manipulation and involves the following steps: 1. Navigate to object A, 2. Grab object A, 3. Navigate to object B, 4. Grab object B, 5. Navigate to recipient location C, and 6. Place objects A and B at recipient location C sequentially. If the task involves placing objects into a container that needs to be open first, the robot must temporarily place one of the objects on a nearby surface before executing the OPEN action. Therefore, we have two versions of plans for navigation and manipulation involving two objects. The plan with the shortest walking distance to complete the task will be determined as the optimal plan.

For tasks involving navigation and manipulation with temporal constraints, a straightforward example is placing object A into object B, and subsequently placing object B into object C. The ultimate goal is to have object A inside object B, which is itself inside object C. This type of task involves stringent temporal constraints because the robot must complete the first sub-task before initiating the second one.

**Prior global information from the environment** To complete the task with prior layout map information, we utilize the pipeline described in Algorithm 1 or Algorithm 3. We identify and extract large immovable furniture in Virtual Home by identifying objects that are neither movable nor grabbable. If the task involves finding an object inside one of these selected pieces of furniture, the agent can navigate directly to it. Otherwise, when the agent is exploring a room, we assume it knows the locations of these pieces of furniture, even if they are not currently visible. Then, the LLM agent can also determine if the target object is inside or near the pre-identified, yet previously unseen, furniture, facilitating further exploration. For task completion with comprehensive environmental information, we employ the ground truth generation pipeline as described in Appendix A.6.

---

**Algorithm 1** Robot's exploration algorithm

---

**Require:** Find the object A
**Ensure:** Reset Virtual Home environment and create a character at a random location
    Get all rooms from the Virtual Home environment
    LLM determines whether the object A is inside these rooms and ranks rooms based on the likelihood of its presence.
    **for** Walk to and explore ranked rooms **do**
        Get visible object lists from 360 degrees exploration
        **if** Object A is inside visible object lists **then**
            Find object A! Finish the task successfully
        **else if** LLM determines whether the object A is inside or obscured by currently visible objects and return any potential large objects **then**
            **for** Walk to and explore potential large objects **do**
                Get visible object lists from 360 degrees exploration
                **if** Object A is inside visible object lists **then**
                    Find object A! Finish the task successfully
                **end if**
            **end for**
        **end if**
    **end for**
    **if** Cannot find object A after exploration **then**
        Failed to finish the task
    **end if**

---

---

**Algorithm 2** Robot's exploration algorithm to finish the navigation task with spatial constraint

---

**Require:** Find the object A which is close to object B
**Ensure:** Reset Virtual Home environment and create a character at a random location
    Get all rooms from the Virtual Home environment
    LLM determines whether both the object A and object B are inside these rooms and ranks rooms based on the likelihood of its presence.
    **for** Walk to and explore ranked rooms **do**
        Get visible object lists from 360 degrees exploration
        **if** Object A is inside visible object lists **then**
            **if** Object B is inside visible object lists and meet criteria (object A, CLOSE, object B) **then**
                Find the object A which is close to object B! Finish the task successfully
            **else**
                Agent walks to object A and get visible object lists from 360 degrees exploration
                **if** Object B is inside visible object lists and meet criteria (object A, CLOSE, object B) **then**
                    Find the object A which is close to object B! Finish the task successfully
                **end if**
            **end if**
        **else if** Object B is inside visible object lists **then**
            Agent walks to object B and get visible object lists from 360 degrees exploration
            **if** Object A is inside visible object lists and meet criteria (object A, CLOSE, object B) **then**
                Find the object A which is close to object B! Finish the task successfully
            **end if**
        **else if** LLM determines whether both the object A and object B are inside or obscured by currently visible objects and return any potential large objects **then**
            **for** Walk to and explore potential large objects **do**
                Get visible object lists from 360 degrees exploration
                **if** Object A is inside visible object lists and object B is inside visible object lists and meet criteria (object A, CLOSE, object B) **then**
                    Find the object A which is close to object B! Finish the task successfully
                **end if**
            **end for**
        **end if**
    **end for**
    **if** Cannot find the object A which is close to object B after exploration **then**
        Failed to finish the task
    **end if**

---

### A.9 EVALUATION METRICS

Given our automated quantitative evaluation pipeline, we utilize various metrics to assess results across different levels of tasks.

**Success Rate** The criteria for determining the success of a sample vary slightly depending on the task. For navigation tasks, a task is deemed successful if it finds the target object. For navigation tasks with spatial constraints, the robot must find a specific object that meets spatial relation criteria. For example, if the task is to find an apple in the fridge but the robot finds an apple on the table, it is not considered as a success. Success rate is calculated as the number of successfully completed cases divided by the total number of cases, with a higher success rate being more favorable.

**Sequence Length** Another key metric is the sequence length, which is the number of action steps required to complete the task. For failed tasks, we use the predefined maximum number of steps as the sequence length. We then average the sequence lengths across all tasks. Shorter sequence lengths are desirable, as they indicate quicker task completion.

**Longest Common Subsequence (LCS) Length** The longest common sequence metric measures the similarity between the agent's exploration actions and the ground truth actions. Typically, in the most successful cases, the length of the longest common sequence matches the length of the ground truth

---

**Algorithm 3** Robot's navigation and manipulation algorithm

---

**Require:** Put object A on object B
**Ensure:** Reset Virtual Home environment and create a character at a random location
   Get all rooms from the Virtual Home environment
   Find object A using Algorithm 1 and record all seen objects information during exploration into memory
   **if** Find object A after exploration **then**
      Agent grabs object A
   **else**
      Failed to finish the task, exit
   **end if**
   **if** Object B is in memory **then**
      Agent walk to object B and place object A on object B. Finish the task successfully
   **else**
      Find object B using Algorithm 1 and regard objects in the memory as a part of visible objects during exploration.
      **if** Find object B **then**
         Agent walk to object B and place object A on object B. Finish the task successfully
      **else**
         Failed to finish the task
      **end if**
   **end if**

---

actions. For a specific task, a longer longest common sequence indicates that the exploration actions are more similar to the ground truth actions.

**Longest Common Subsequence Ratio** Using longest common sequence as evaluation metric can be biased as it favors complex tasks with longer action lists. To ensure fair comparisons between tasks of varying action lengths, we introduce the longest common subsequence ratio, which is calculated by dividing LCS length with ground truth (GT) length. The optimal value for this ratio is 1.

**Moving Distance** We also introduce an additional evaluation metric: moving distance for successful cases. For these cases, a shorter moving distance indicates a more efficient moving path. The total distance is calculated incrementally as the agent moves to new positions.

### A.10 ADDITIONAL EXPERIMENT RESULTS

#### A.10.1 TASK PLANNING RESULTS WITH CHAIN OF THOUGHT

Chain of Thought (CoT) is frequently employed in large language models (LLMs) to enhance their reasoning capabilities (Wei et al., 2022). Notably, several zero-shot CoT methods (Kojima et al., 2022) have achieved fairly good performance across diverse task domains. For instance, merely appending "Let's think step by step" can improve results consistently.

As shown in Table 6, navigation tasks with CoT achieves slightly better success rates and LCS ratio. Moreover, CoT enhances the interpretability of the planning process, which is crucial for household robots and potentially enhancing the human-robot interaction experience.

Two prompts with Chain of Thought (CoT) reasoning are presented in Table 14 and Table 15. Additionally, GPT-4 provides reasonable responses based on these prompts. Two demonstration examples are shown in Table 16 and Table 17.

Prompt engineering is crucial for inspiring the best performance in large language models. We made sufficient efforts to evaluate and choose the most effective prompt for our final use. While it may not be the absolute best, since it's impossible to definitively determine the best prompt, we can confidently say that our chosen prompt is sufficient for our tasks. Overall, we kept our prompts simple, clear, and with in-context examples, following standard prompt design guidelines.

Table 6: Comparison for results with/without Chain of Thought

|  | Success Rate | Seq Length | Longest Common Seq (Ratio) | Moving Distance |
|---|---|---|---|---|
| Navigation | 79.26% | 6.77 | 1.59 (78.74%) | 14.10 |
| Navigation + *CoT* | 79.73% | 6.78 | 1.62 (80.21%) | 14.38 |

Table 7: Comparison between zero-shot prompt and few-shot prompt

| Task | Success Rate | Seq Length | LCS (Ratio) | Moving Distance |
|---|---|---|---|---|
| Zero-shot prompt + Navi | 79.26% | 6.77 | 1.59 (78.74%) | 14.10 |
| Few-shot prompt + Navi | 77.81% | 7.08 | 1.57 (77.61%) | 14.08 |
| Zero-shot prompt + Navi + Relation | 62.61% | 9.20 | 1.78 (88.75%) | 12.45 |
| Few-shot prompt + Navi + Relation | 62.59% | 9.12 | 1.77 (88.63%) | 11.97 |

### A.10.2 TASK PLANNING RESULTS WITH FEW-SHOT PROMPTING

We conducted a comprehensive evaluation of the performance of few-shot prompting and provided a comparison with zero-shot prompting, as shown in the Table!7. Our findings indicate that the performance difference between few-shot and zero-shot prompting is negligible. For the sake of simplicity, we retained the zero-shot prompt version.

### A.10.3 OBJECT DETECTION ANALYSIS IN VIRTUAL HOME

We aimed to incorporate visual input to propose a multi-modal benchmark for embodied planning, recognizing that in real-world scenarios, robots primarily perceive information through visual data. To this end, we evaluated Grounding-DINO (Liu et al., 2023), one of the state-of-the-art object detection methods, as a perception module, along with two different vision-language models: GPT-4V, and LLaVA-7B.

To evaluate the object detection capabilities of these models in Virtual Home, we simulated the 360-degree perspective used in our search algorithm across all 50 environments, each containing 4 rooms. This process generated a total of 1,155 images after filtering out views with no visible objects.

For Grounding-Dino we feed the list of assets in Virtual Home separated with a '.', to detect each object as a particular class. The list of assets can be found in Table 19. For the prompt used for GPT-4V and LLaVA see Table 18.

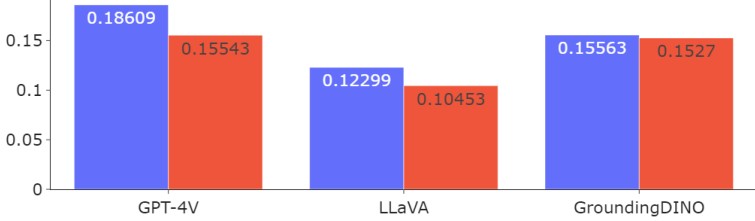

Figure 20: This figure shows the improvement of the mean F1 score before (red) and after (blue) applying the embedding matching with cosine similarity technique. The improvement goes around 2% to 3% among VLMs.

To mitigate the problem where the output of the VLMs don't match the asset name in Virtual Home, due to the zero-shot setting, we followed the same strategy as (Zhao et al., 2024) to match word-pairs using sentence-BERT (Reimers & Gurevych, 2019). We calculate the word embedding for each asset

in the output, as well as each asset from Virtual Home, and then select the most similar pair using cosine similarity. We defined a similarity threshold of 0.8 to decide if we match these words. This way we make the models output to be compatible with Virtual Home, e.g. "dish washing liquid" to "dishwashingliquid". General improvement for each model is shown at Figure 20.

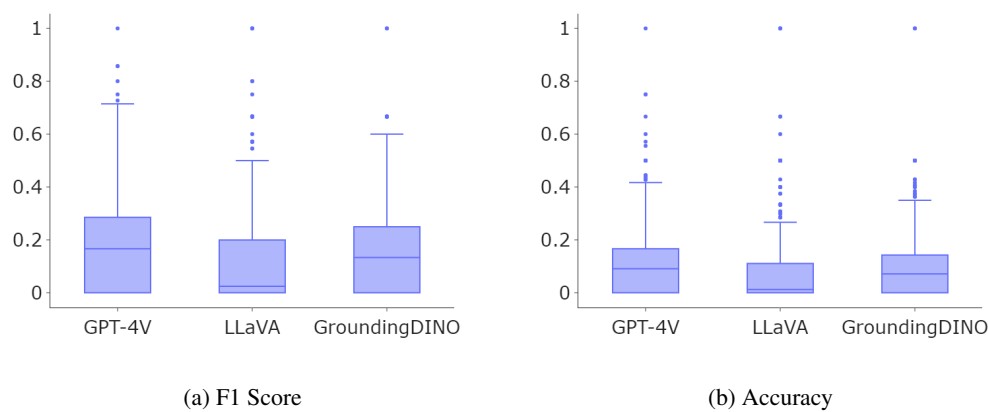

|                  (a) F1 Score                  |                 (b) Accuracy                  |

Figure 21: Evaluation results for the object detection in Virtual Home images. We observe similar performances between GPT-4V and GroundingDINO, and a poor performance for LLaVA.

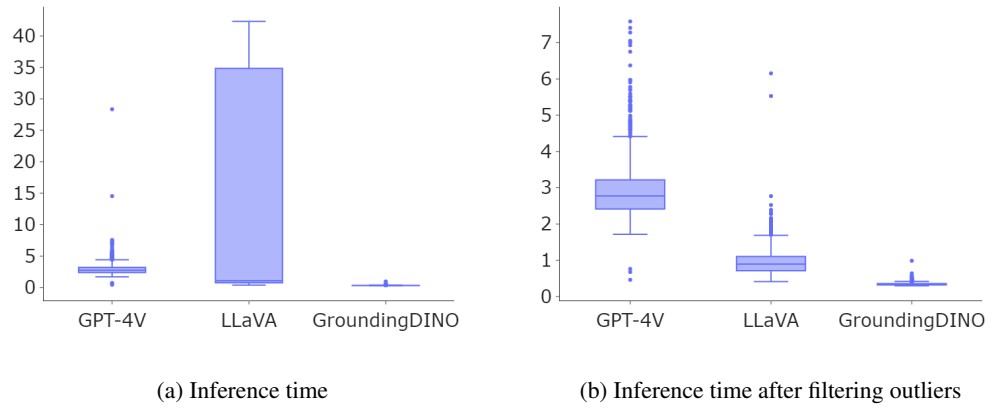

|               (a) Inference time               |      (b) Inference time after filtering outliers      |

Figure 22: Figure 22a shows a high peak inference on LLaVA. As explained in Figure 23, this comes because of the model repeating the output. For a clearer comparison, we filtered these outlier cases in Figure 22b. We see GroundingDINO has the lowest inference, much faster than VLMs, being 3x faster than LLaVA (filtered results). Since GPT-4V is employed via an API, its inference time is incomparable to that of the other two methods.

To evaluate the correctness of the output of these visual models, we calculated the F1 score and the accuracy using the object list retrieved by Virtual Home as the ground truth for each image. We also evaluated the inference response to measure the performance gain with higher computation cost for each model. Results showed in Figure 21 demonstrate that there is not a big performance gain by using current state-of-the-art VLM models against an object detection model as GroundingDINO. More than that, we can see in Figure 22 that using VLMs increase the inference time significantly. This makes it unfeasible for our embodied task planning method, and its one of the reasons why we decide to use the Virtual Home API instead. Since Habitat does not provide a perception module API like Virtual Home, we utilized Grounding-DINO as the perception module for the LLM agent within the Habitat virtual environment.

There are a couple reasons why these object detection models are not reliable for our method. First, the images provided by Virtual Home are in low resolution, what leads to ambiguous interpretation

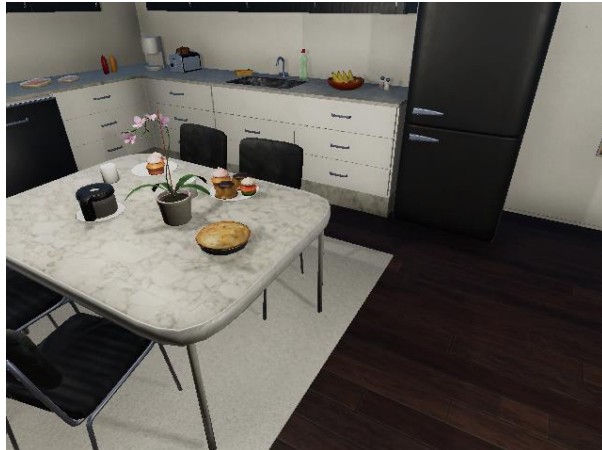

**LLaVA Output**

- **False negative:** ['toothbrush', 'hairproduct', 'toothpaste', 'plate', 'clothespile', 'candle', 'bookshelf', 'creamybuns', 'creamybuns', 'chips', 'chocolatesyrup']

- **False positive:** ['bottle', 'bottle', 'bottle', 'bottle', 'bottle', 'bottle', 'bottle', 'bottle', 'bottle', 'bottle', 'bottle', 'bottle', 'bottle', 'bottle', 'bottle', 'bottle', 'bottle', 'bottle', 'bottle',....]

Figure 23: For some images, LLaVA will steadily repeat the same object in the output until the maximum number of tokens is reached. Particularly, 356 out of 1,155 (30.82%) images evaluated presented this repeating problem with LLaVA, which shows a poor instruction following ability with LLAVA-7B model. This behavior causes larger inferences as shown in Figure 22a.

**GPT-4V Output**

- **True positive:** ['chair', 'chair', 'chair', 'pie', 'plate', 'plate', 'faucet']

- **False negative:** ['toaster', 'breadslice', 'breadslice', 'coffeemaker', 'condimentbottle', 'condimentbottle', 'condimentshaker', 'coffeepot', 'orchid', 'bananas', 'dishbowl', 'mincedmeat', 'cutlets', 'dishwashingliquid', 'washingsponge', 'kitchencabinet', 'kitchencabinet', 'kitchencabinet', 'kitchencabinet', 'kitchencabinet', 'kitchencabinet', 'kitchencounter', 'dishwasher',]

- **False positive:** ['refrigerator', 'table', 'chair', 'plant', 'cupcake', 'cupcake', 'cupcake', 'coffee pot', 'cabinet', 'cabinet', 'cabinet', 'cabinet', 'cabinet', 'cabinet', 'cabinet', 'cabinet']

Figure 24: In this example with GPT-4V we see objects that are visible are not detected by Virtual Home, therefore being categorized as false positives, e.g. cupcakes, refrigerator, table, etc. Other objects like the kitchen cabinets or the orchid fail the embedding-matching with their corresponding model outputs, cabinet and plant.

of the objects in the image even for humans, particularly the smaller ones. As shown in Figure 22 and Figure 23, inference times can be significant, and as high as 30 seconds per inference. This issue represents a significant bottleneck in the current stage of integrating multi-modality into our LLM agent.

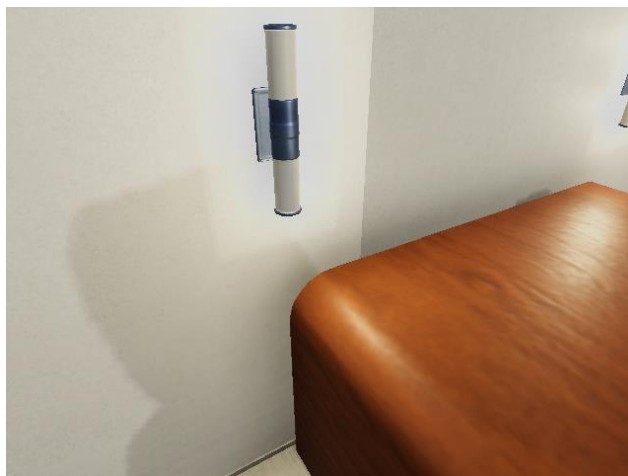

**GPT-4V Output**

- **False negative:** ["book", "book", "door"]

- **False_positive:** ["wall light", "wall light", "bed"]

Figure 25: In this example with GPT-4V, we see objects that are visible in the image not being included in Virtual Home's ground truth object list, therefore being categorized as false positives.

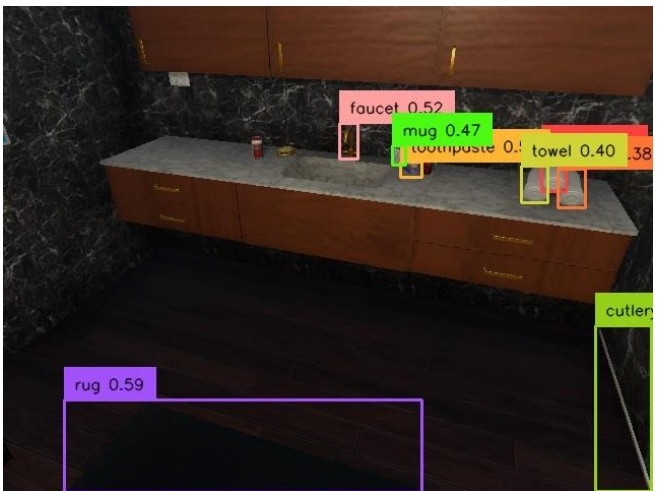

**Grounding Dino Output**

- **True positive:** ['towel', 'faucet', 'towel', 'toothpaste', 'towel', 'barsoap']

- **False negative:** ['stall', 'sink', 'closet', 'toothbrush', 'deodorant', 'facecream', 'painkillers', 'wallpictureframe', 'powersocket']

- **False positive:** ['rug', 'mug', 'cutleryfork']

Figure 26: GroundingDINO provides bounding boxes along with their corresponding confidence scores, adding a layer of interpretability that is often lacking with VLMs. However, it still requires us to specify the classes we want to detect as input to achieve good detection accuracy. Additionally, it struggles with low-resolution images from Virtual Home.

Other reason is our current way to match words using embedding. Some outputs, like as in Figure 24, show that there is still margin of improvement in how we align the output of our model and the Virtual Home environment.

There are also cases where the Virtual Home API provide objects not visible in the image, or vice versa, don't include obvious visible objects in the list, as shown in Figure 25. Even with a perfect detection model, we cannot fully rely on it due to these inherent errors from Virtual Home.

Table 8: Task generation prompt for navigation & manipulation + multiple objects

I will give a task [task] to put object 1 and object 2 inside or on the object 3. Please make a decision whether it makes sense in the realistic world. Besides, there are rules that needs to be considered. If the task does not meet the rules, it is still considered as unreasonable.
1. The object 1 needs to be grabbable reasonably for a human.
2. The object 2 needs to be grabbable reasonable for a human as well.
3. In general, the object 1 and object 2 together needs to be smaller than the object 3.
4. The object 1 and object 2 are reasonably related. For instance, a apple and a banana is related as they are both fruits. A apple and a knife are also related, as we can use a knife to cut a apple.
5. Based on common sense, the task needs to make sense in realistic world. For instance, putting desk and table into the frying pan is not a reasonable task.
Following are examples:
task: Find an apple and a knife, then put them into the box
conclusion: Reasonable
task: Can you find knife and the apple, and put them into a tray for later use
conclusion: Reasonable
task: Please put the bowl and the cellphone into the sink
conclusion: Unreasonable
task: Locate the toothbrush and the condiment shaker, then place them on the bed
conclusion: Unreasonable
task: Find a computer mouse and a peach, then place them on the bench
conclusion: Unreasonable
Please only output 'Reasonable' or 'Unreasonable' as output. Do not output thinking process, or extra words such as 'conclusion' or API_key

We didn't directly test the bounding box capabilities of VLM, that we have with Grounding-DINO, as seen in Figure 26, but zero-shot outputs showed it might hallucinate these values. Virtual Home assets don't have specific properties on the class names, like colors, shapes, etc. For future work, it still needs study to evaluate how these models perform detecting more specific objects, e.g. big red apple.

### A.11 ACCESSIBILITY

**URL to code and dataset:** The code and benchmark data samples are included in supplementary material and will be publicly released once the paper is accepted. We commit to maintaining both the data and the code. Our data are intended for academic use.

**Benchmark dataset meta data** Our benchmark dataset is primarily composed of four key components: task descriptions, task completion criteria, environment information, and ground truth action plans.

**Author statement:** We acknowledge full responsibility for any violation of rights that may occur during data collection or related activities, and we will take appropriate action as needed, such as removing any data implicated in such issues.

**License:** ET-Plan-Bench is released under CC BY-NC 4.0 License.

### A.12 PROMPTS

Table 9: Task generation prompt for navigation & manipulation + temporal dependency

I will give a task [task] to put object 1 inside object 2 and then put them on or inside the object 3. Please make a decision whether it makes sense in the realistic world. Besides, there are rules that needs to be considered. If the task does not meet the rules, it is still considered as unreasonable
1. The object 1 needs to be grabbable reasonably for a human.
2. The object 2 needs to be grabbable reasonable for a human as well.
3. The object 1 needs to be smaller than the object 2.
3. The object 1 and object 2 together needs to be smaller than the object 3.
4. There needs to be a reason why the object 1 needs to be put inside object 2 before being placed on the object 3. For instance, a apple is placed into a small box before placed on the kitchen counter, because the counter top can look more organized this way.
5. Based on common sense, the task needs to make sense in realistic world. For instance, place telephone into the frying pan and then place them into the sink is not a reasonable task.
Following are examples:
task: Please put book into box and then put them on the desk
conclusion: Reasonable
task: Can you put the knife into the box, and place them together on the kitchen counter
conclusion: Reasonable
task: Please put the cellphone into the bowl and put bowl on the table
conclusion: Unreasonable
task: Locate the toothbrush into condiment shaker, then place them on the floor
conclusion: Unreasonable
Please only output 'Reasonable' or 'Unreasonable' as output. Do not output thinking process, or extra words such as 'conclusion' or API_key

Table 10: Generated Question-and-Answer (QA) pairs and executable action plans

**Task:** Where is the toilet?
**Prompt (Question):** Determine which room may contain the object toilet, and the room list is ['bathroom', 'bedroom', 'kitchen', 'livingroom']. Please ranking these rooms based on the possibility and only output a Python-style list. The number of output rooms should be the same as the number of rooms in the original room list. Please do not output the answer like 'As an AI language model, I don not have the ability to physically determine the location of objects or bring them to you.' Please do not output the thinking process.
**GPT4 Response (Answer):** ['bathroom', 'bedroom', 'livingroom', 'kitchen']
**Prompt (Question):** Currently, the robot is in the room bathroom and can see objects ['hairproduct', 'mug', 'plate', 'character', 'cpuscreen', 'candle', 'wallshelf', 'stall', 'desk', 'facecream', 'toothpaste', 'bathroom', 'painkillers', 'bathtub', 'door', 'faucet', 'towel', 'mouse', 'bathroomcounter', 'wallpictureframe', 'washingmachine', 'chair', 'bedroom', 'barsoap']. Since the target object could be obscured by some larger visible objects, the robot needs to explore the room further. Is this possible the object toilet is located inside or nearby one of these visible objects? If yes, please only output these possbile visible objects in a Python-style list and in the order of possibility, if not, please only output None. Please do not output the thinking process.
**GPT4 Response (Answer):** ["stall", "bathtub"]
**Executable Action Plans:**
1. walk bathroom
2. walk stall
3. walk toilet

Table 11: Task classification prompt

Please classify the task: [task name]. Total classes are:
1. simple navigation: this class involves navigating something or giving something to a person.
2. constrained simple navigation: this class involves navigating or finding something with spatial constraints.
3. pick and place: the class involves navigating and picking the first object, and then placing the first object to another place, rather than a person.
4. constrained pick and place: this class involves navigating and picking the first object with spatial constraints, and then placing the first object to another place with spatial constraints.
5. pick two objects and place jointly: this class involves picking two objects and placing them to another place, rather than a person.
Here are some examples:
1. Task: Can you pick up an apple for me? Class: simple navigation.
2. Task: Give me an apple in the fridge. Class: constrained simple navigation.
3. Task: Can you please place the clothespants on the kitchen table? Class: pick and place.
4. Task: Please locate the breadslice inside the kitchen and place it on the kitchentable also inside the kitchen. Class: constrained pick and place.
5. Task: Find the cereal and the keyboard, then put them into the drawer for storage. Class: pick two objects and place jointly.
6. Task: Can you point me to the microwave? Class: simple navigation.
7. Task: Can you close the curtains? Class: simple navigation.
8. Task: Can you bring me the mug on the desk? Class: constrained simple navigation.
9. Task: Can you get the barsoap from inside the bathroom? Class: constrained simple navigation.
10. Task: Can you find the condiment shaker and put it in the garbage can? Class: pick and place.
11. Task: Can you place the peach close to the dishbowl on the bench on the rug? Class: constrained pick and place.
12. Task: Can you please find the crackers facing the TV and put them on the chair inside the bedroom? Class: constrained pick and place.
13. Task: Find the cereal and the condiment shaker, then put them on the stove for preparing breakfast. Class: pick two objects and place jointly.
Please only output the class name at the last line of the answer. Let's think step by step.

Table 12: Navigation ranking rooms prompt

Determine which room may contain the object [object name], and the room list is [all rooms list]. Please rank these rooms based on the possibility and only output a Python-style list. The number of output rooms should be the same as the number of rooms in the original room list. Please do not output the answer like 'As an AI language model, I don not have the ability to physically determine the location of objects or bring them to you.' Please do not output the thinking process.

Table 13: Navigation large objects selection prompt

Currently, the robot is in the room [room name] and can see objects [visible objects list]. Since the target object could be obscured by some larger visible objects, the robot needs to explore the room further. Is this possible the object [object name] is located inside or nearby one of these visible objects? If yes, please only output these possible visible objects in a Python-style list and in the order of possibility, if not, please only output None. Please do not output the thinking process.

Table 14: Navigation ranking rooms prompt with CoT

Determine which room may contain the object [object name], and the room list is [all rooms list]. Please ranking these rooms based on the possibility and only output a Python-style list. The number of output rooms should be the same as the number of rooms in the original room list. Please do not output the answer like 'As an AI language model, I do not have the ability to physically determine the location of objects or bring them to you.' Please output the thinking process and the final answer. The final answer should be in the last line and should follow the format mentioned above. Please do not output useless text in the line of final answer, such as 'The final answer:', 'The final answer is', and so on. Let's think step by step.

Table 15: Navigation large objects selection prompt with CoT

Currently, the robot is in the room [room name] and can see objects [visible objects list]. Since the target object could be obscured by some larger visible objects, the robot needs to explore the room further. Is this possible the object [object name] is located inside or nearby one of these visible objects? If yes, please only output these possible visible objects in a Python-style list and in the order of possibility, if not, please only output None. Please output the thinking process and the final answer. The final answer should be in the last line and should follow the format mentioned above. Please do not output useless text in the line of final answer, such as 'The final answer:', 'The final answer is', and so on. **Let's think step by step.**

Table 16: Navigation ranking rooms prompt-response demo

**Prompt** Determine which room may contain the object whippedcream, and the room list is ['bathroom', 'bedroom', 'kitchen', 'livingroom']. Please ranking these rooms based on the possibility and only output a Python-style list. The number of output rooms should be the same as the number of rooms in the original room list. Please do not output the answer like 'As an AI language model, I don not have the ability to physically determine the location of objects or bring them to you.' Please output the thinking process and the final answer. The final answer should be in the last line and should follow the format mentioned above. Please do not output useless text in the line of final answer, such as 'The final answer:', 'The final answer is', and so on. Let's think step by step.

**GPT4 Response** To determine which room might contain whipped cream among the options provided 'bathroom', 'bedroom', 'kitchen', 'livingroom', we can analyze typical usage and storage behavior for whipped cream:

1. **Kitchen**: Whipped cream is commonly used in cooking and food preparation. It is typically stored in a refrigerator which is mostly located in the kitchen. This makes the kitchen the most likely place to find whipped cream.

2. **Living Room**: While not typical for storing food items, sometimes during gatherings or parties, food items including whipped cream might be temporarily placed in the living room for accessibility and convenience. However, this is less likely than finding it in the kitchen.

3. **Bedroom**: It's rather uncommon to store food in the bedroom, especially whipped cream, which requires refrigeration. However, without totally disregarding the possibility, whipped cream might be found here in less conventional situations, perhaps during a special occasion like breakfast in bed.

4. **Bathroom**: It is highly unlikely to find whipped cream stored in a bathroom due to the need for refrigeration and the typical use of bathrooms for hygiene products and medications. With this analysis, we can rank the rooms by likelihood of containing whipped cream: ['kitchen', 'livingroom', 'bedroom', 'bathroom']

Table 17: Navigation large objects selection prompt-response demo

**Prompt** Currently, the robot is in the room bathroom and can see objects ['wallshelf', 'faucet', 'character', 'painkillers', 'chair', 'candle', 'plate', 'mouse', 'bedroom', 'bathtub', 'bathroom', 'washingmachine', 'towel', 'door', 'hairproduct', 'facecream', 'wallpictureframe', 'stall', 'toothpaste', 'bathroomcounter', 'cpuscreen', 'barsoap', 'mug', 'desk']. Since the target object could be obscured by some larger visible objects, the robot needs to explore the room further. Is this possible the object toilet is located inside or nearby one of these visible objects? If yes, please only output these possible visible objects in a Python-style list and in the order of possibility, if not, please only output None. Please output the thinking process and the final answer. The final answer should be in the last line and should follow the format mentioned above. Please do not output useless text in the line of final answer, such as 'The final answer:', 'The final answer is', and so on. Let's think step by step.

**GPT4 Response** To address the prompt, the task is to determine if the object "toilet" can be found inside or close to one of the visible objects listed. Here is the breakdown of our thought process:
1. **Analyzing Visible Objects**: We must first identify objects from the list that are commonly found in a bathroom and large enough to potentially obscure or be in close proximity to a toilet.
2. **Common Bathroom Objects**: From the list, objects like 'bathtub', 'bathroomcounter', 'stall', and 'door' immediately stand out as potentially relevant. These objects are typical in bathrooms and are large enough to either hide a toilet from view or be positioned near it.
3. **Exploring the Selected Objects**: - **Bathtub**: While it is a large fixture in a bathroom next to which toilets are often positioned, assignment obstruction is less likely but proximity is possible. - **Bathroomcounter**: Typically houses sinks and associated elements but isn't usually positioned very close to toilets. - **Stall**: Public or larger bathrooms might have toilets inside stalls, making this a likely candidate if the robot is in such a setting. - **Door**: As the entry to potentially secluded toilet sections or individual toilet rooms (especially in spaces like public restrooms), this could be relevant.
4. **Practical Considerations**: The context provided by the visibility in a bathroom setting (or similarly structured spaces) implies either partial obstruction by these objects or close proximity to them.
5. **Order of Possibility**: - **Stall**: Very high likelihood if the setup includes individual stalls typically containing toilets. - **Door**: High possibility if it serves as an entryway to an individual toilet room or section. - **Bathtub**: Moderate possibility depending on the bathroom layout; often positioned close to a toilet. - **Bathroomcounter**: Lower likelihood compared to others, usually houses sink and storage spaces.
By synthesizing these observations, one can make an informed hypothesis about where the toilet might be located or obscured. ['stall', 'door', 'bathtub']

Table 18: Object detection analysis prompt

**Prompt:** Your task is to detect all the objects in the given image(s), and return them in a list. If there are no valid reasons why an object is in the image, don't include it in the response. Please do not output something like: 'I'm sorry, but I cannot assist with requests involving the analysis or processing of images or videos.' In the last line of your answer, output only the name of the detected objects as classes in a Python-style list. For example: ['apple', 'banana', 'peach']. If there are multiple objects of the same class, add them individually. For example, if there are 2 apples and 1 plate in the image, output: ['apple', 'apple', 'plate'] and not ['apples', 'plate']

Table 19: Virtual Home object assets

['rug', 'chair', 'towelrack', 'candle', 'box', 'ceilinglamp', 'wallshelf', 'faucet', 'closetdrawer', 'walllamp', 'doorjamb', 'tablelamp', 'coffeetable', 'tvstand', 'garbagecan', 'desk', 'kitchentable', 'curtains', 'bench', 'toilet', 'bookshelf', 'slippers', 'clothesshirt', 'clothespile', 'remotecontrol', 'keyboard', 'mouse', 'cellphone', 'radio', 'clock', 'wallphone', 'microwave', 'lightswitch', 'ceilingfan', 'stovefan', 'speaker', 'computer', 'powersocket', 'amplifier', 'cpuscreen', 'bananas', 'cupcake', 'whippedcream', 'chips', 'crackers', 'candybar', 'breadslice', 'lime', 'chocolatesyrup', 'creamybuns', 'peach', 'plum', 'pie', 'cereal', 'bellpepper', 'salmon', 'waterglass', 'oventray', 'cutleryknife', 'mug', 'dishwashingliquid', 'condimentbottle', 'fryingpan', 'cutleryfork', 'washingsponge', 'wineglass', 'dishbowl', 'plate', 'condimentshaker', 'coffeepot', 'cookingpot', 'pancake', 'milkshake', 'chicken', 'juice', 'cutlets', 'hairproduct', 'barsoap', 'towel', 'toothpaste', 'facecream', 'toothbrush', 'toiletpaper', 'deodorant', 'perfume', 'folder', 'book', 'paper', 'crayons', 'notes', 'magazine', 'papertray', 'wallpictureframe', 'orchid', 'toy', 'boardgame', 'guitar', 'painkillers', 'box', 'closet', 'sink', 'cabinet', 'kitchencabinet', 'bathroomcabinet', 'sofa', 'bed', 'bathroomcounter', 'clothespants', 'hanger', 'pillow', 'tv', 'mousemat', 'washingmachine', 'stove', 'fridge', 'printer', 'apple', 'pear', 'salad', 'sundae', 'mincedmeat', 'pie', 'alcohol', 'milk', 'pudding', 'cuttingboard', 'knifeblock', 'toaster', 'coffeemaker', 'microwave']

