# OpenReview forum: "ET-Plan-Bench: Embodied Task-level Planning Benchmark Towards Spatial-Temporal Cognition with Foundation Models"
_ICLR.cc/2025/Conference — Submitted to ICLR 2025_

### Official Review · Reviewer_VG5n · 2024-11-02

**Soundness:** 2
**Presentation:** 1
**Contribution:** 3
**Rating:** 5
**Confidence:** 4

**Summary:**

The authors present a new embodied task planning benchmark that uses language models to generate tasks.  They also present a language baseline agent for this task.

**Strengths:**

+ The work here appears large and comprehensive.
+ Exploring the extent to which language models can solve embodied tasks is extremely relevant.
+ Designing systems which can procedurally generate new data is useful.

**Weaknesses:**

- Minor: two sentences from the abstract are nearly copied to the beginning of the introduction this is unnecessary and should be avoided.
- Major: At the moment, the presentation Is quite unclear.  While the overall goal, using LLMs to both generate tasks and solve embodied AI problems is apparent, the method, the primary assumptions and overall system are hard to understand.
- Major: It's not clear what using the LLM in task generation accomplishes.  Is the goal just to provide common sense reasoning over the larger task search space?  Otherwise, it seems like these tasks can be generated fairly mechanically as others have done in the past.  To be clear my complaint is not that the LLM serves no purpose, my complaint is that it is not well described (again a clarity issue), and that it's hard to tell how much this adds.
- Medium: It says that humans evaluated these tasks based on "reasonableness" which seems highly subjective and difficult to quantify.

**Questions:**

- What is the action space and observation space in this setting?
- How does the language model interact with the simulator?
- Is there any way to compare tasks generated with the LLM to tasks that are sampled some other way (randomly sampling objects and compatible containers for example)?  Is there any way to tell how much we are gaining by using the LLM in the data generation process?
- Are the LLM agents taking in the rendered images?  If so, shouldn't we be referring to these as VLMs to make this more clear?

My review for the moment is "5: marginally below the acceptance threshold" but I would really like to improve my score here.  There is clearly a lot of effort and good work that went into this paper, but at the moment, a lot of important components of this system are not clear.  To meet the bar for acceptance, it is necessary to be able to explain how an LLM interfaces with one of these 3D simulators to an audience that is familiar with RL and LLMs individually, but has never seen them combined, and this information should be clearly stated in the main text (the introduction should even give a rough sense of this) and not buried deep in the appendices.

---

> ### Author Response · Authors · 2024-11-22
> **Rebuttal by Authors**
>
> We thank the reviewer for their insightful comments and your encouraging feedback. Please find our overall concise response to your main questions below.
> 1. Comments: Minor: two sentences from the abstract are nearly copied to the beginning of the introduction this is unnecessary and should be avoided.
>
> Response: Thank you for your suggestions. We have addressed and avoided this issue in our revision.
>
> 2. Comments: Major: At the moment, the presentation Is quite unclear. While the overall goal, using LLMs to both generate tasks and solve embodied AI problems is apparent, the method, the primary assumptions and overall system are hard to understand.
>
> Response: Thank you for your comments. We have revised our paper to the best of our ability to improve clarity.
>
> 3. Comments: Major: It's not clear what using the LLM in task generation accomplishes. Is the goal just to provide common sense reasoning over the larger task search space? Otherwise, it seems like these tasks can be generated fairly mechanically as others have done in the past. To be clear my complaint is not that the LLM serves no purpose, my complaint is that it is not well described (again a clarity issue), and that it's hard to tell how much this adds.
>
> Response: Thanks for your suggestion. We have included an example to clarify the use of the LLM in task generation: "After retrieving the relevant information from the environment, we shortlist all possible item combinations that meet the requirements of the commission template. For instance, the task of placing object A into container B necessitates retrieving both object A and container B from the environment. We subsequently employ LLMs to generate tasks adhere to principles of logical common sense and are rephrased in diverse ways to increase the diversity of descriptions. The LLMs receive both the environmental information and the task template as part of the prompt, which also includes rules and examples to ensure the tasks generated are realistic." The basic idea involves first defining a task template and then filling it with object names in the template's blank spaces. The next step is to use a LLM to filter out unreasonable tasks, thereby ensuring the diversity of reasonable task descriptions. Two example prompts of the LLM are shown in Table 8 and Table 9 in the Appendix.
>
> 4. Comments: Medium: It says that humans evaluated these tasks based on "reasonableness" which seems highly subjective and difficult to quantify.
>
> Response: We respectfully wish to clarify that there may have been some misunderstandings regarding our paper. In Section 3.2, "we randomly selected 5 simulation environments and chose 50 tasks from each environment for human evaluation, and our findings indicate that approximately 4% to 8% of the tasks were deemed unreasonable." We agree that human evaluation is inherently subjective; however, we do not rely on it to filter out unreasonable tasks. Instead, we employ human evaluation to ensure that the majority of the tasks we generate appear reasonable; in other words, they seem common to humans. Although a small number of tasks were deemed unreasonable or unusual based on our human evaluation, they can still be successfully executed in the VirtualHome simulator. This is because we systematically filter out unreasonable tasks using LLMs and simulator verification during our task generation process. One example of an unreasonable task by human evaluation we mentioned in the paper is "Find the face cream tube and put it into the folder." While some might consider this task reasonable or common in practice, we classify it as unusual due to our relatively strict human evaluation criteria.  Other examples that were deemed unreasonable or unusual based on human evaluation are as follows: "Your task is to retrieve cellphone and deposit it into box, subsequently transferring both into garbagecan.", "Find a bunch of bananas and put them into the basket located by the toilet." and "Find the toothbrush and place it in the coffeemaker's cup."
>
> 5. Comments: What is the action space and observation space in this setting?
>
> Response: The action space in this setting includes WALK, OPEN, CLOSE, GRAB, PUT, and PUTIN, while the observation space comprises a 360-degree scan. These concepts were introduced in Section 4.1.

---

> > ### Author Response · Authors · 2024-11-22
> > **Rebuttal by Authors (second part)**
> >
> > 6. Comments: How does the language model interact with the simulator?
> >
> > Response: LLMs have the capability to decompose tasks into several subgoals. For example, the task of picking up object A and placing it on object B can be broken down into several sequential steps: 1. walking to object A, 2. grabbing object A, 3. walking to object B, and 4. placing object A on object B. Additionally, an LLM can determine which room is most likely to contain the target object and whether the target object could be obscured by some larger objects. In the VirtualHome environment, each subgoal involves navigating to a specific object or location, such as the center of a room. The RL-based path planning method in VirtualHome is then employed to control the robot's movements. The simulator provides feedback to the LLM agent, indicating whether the robot has reached the specified location or successfully grabbed the target object. Based on this feedback, the LLM agent will re-plan its decomposition and interact with the simulator iteratively until the task is completed or the maximum number of attempts is reached.
> >
> > 7. Comments: Is there any way to compare tasks generated with the LLM to tasks that are sampled some other way (randomly sampling objects and compatible containers for example)? Is there any way to tell how much we are gaining by using the LLM in the data generation process?
> >
> > Response: We would like to kindly clarify any potential misunderstandings regarding our paper. We did not use LLM to generate tasks directly. Instead, we employed the LLM to assess whether the tasks were reasonable. For instance, we evaluated whether the sampling object could be placed inside a compatible container in real-world scenarios. Additionally, we used the LLM to rephrase task descriptions to enhance their diversity.
> >
> > 8. Comments: Are the LLM agents taking in the rendered images? If so, shouldn't we be referring to these as VLMs to make this more clear?
> >
> > Response: The entire LLM agent pipeline processes images. In the paper, we utilize the perception modules: the perception API in VirtualHome and GroundingDINO in Habitat, to extract object information from the images. This object information is then fed into the LLM to facilitate decision-making. Consequently, we do not employ VLMs at the current stage, as VLM agents present greater challenges that we intend to explore in future work.
> >
> > 9. Comments: Explain how an LLM interfaces with one of these 3D simulators to an audience that is familiar with RL and LLMs individually, but has never seen them combined, and this information should be clearly stated in the main text (the introduction should even give a rough sense of this) and not buried deep in the appendices.
> >
> > Response: Following your suggestion, we have incorporated this into the Section Introduction in the revised version: "In the context of embodied planning, LLMs focus on the decomposition of complex tasks. The LLM acts both as a translator and a mediator. Each sub-task, such as navigating to a specific location or grabbing a particular object, can be executed using advanced Reinforcement Learning (RL) methods."

---

> > > ### Author Response · Authors · 2024-12-02
> > > **Looking forward to hearing from you soon**
> > >
> > > Dear reviewer,
> > >
> > > We want to follow up on our previous response and the improvements we've made based on your insightful review. As the rebuttal period nears its end, we would like to ensure that we have addressed all your concerns. Your feedback is invaluable to us, and we welcome any additional comments you might have.
> > >
> > > Thank you for your time and effort in helping us enhance our work.
> > >
> > > We are looking forward to your response.

---

### Official Review · Reviewer_sWrr · 2024-11-03

**Soundness:** 2
**Presentation:** 2
**Contribution:** 2
**Rating:** 5
**Confidence:** 5

**Summary:**

This paper presents ET-Plan-BENCH, a benchmark for evaluating large language models (LLMs) on embodied task planning. The benchmark includes navigation and manipulation tasks with varying spatial and temporal constraints, implemented in Virtual Home and Habitat simulators. The main contributions are: (1) A task generation pipeline using LLMs, (2) A benchmark focusing on spatial-temporal cognition, and (3) Baseline results comparing different LLM agents. The paper evaluates several metrics including success rate, action sequence length, and moving distance.

**Strengths:**

The paper presents an attempt to systematically evaluate LLMs on embodied planning tasks.

The paper proposes a method to use LLM to generate navigation and manipulation tasks from two simulated environments, Virtual-Home and Habitat.

The paper evaluates LLM agents including GPT-4 and open-source LLMs, LLAMA and Mixtral 8x7B using the proposed benchmark. Results show that a supervised fine-tuned (SFT) small-size LLAMA-7B model using the benchmark EQA data can match the state-of-the-art closed-source model GPT-4.

**Weaknesses:**

Limited novelty in contributions: Task generation using LLMs has been explored extensively in prior work The benchmark tasks are relatively simple and constrained

Significant limitations in the experimental setup: The simulators (Virtual Home and Habitat) are quite restricted in terms of task complexity and environmental interactions Agents do not have true perception - they receive text descriptions rather than visual input. The spatial and temporal constraints are quite basic

Lack of rigorous analysis and insights: The LLM agent comparison results are presented as they are without novel insights No clear framework for systematic task generation and evaluation Limited discussion of how the findings could improve embodied AI systems

Compared to recent work like "Embodied Agent Interface: Benchmarking LLMs for Embodied Decision Making https://arxiv.org/abs/2410.07166" (note that this is not considered part of related work as the submission is concurrent), this paper: Lacks a principled framework for formalizing different types of embodied tasks Does not provide fine-grained error analysis Misses systematic evaluation of different LLM capabilities

The writing is quite poor. It did not explain the setup clearly. I have to find information in the appendix to continue reading.

**Questions:**

How does the task generation ensure coverage of meaningful spatial-temporal constraints? It seems that only limited constraints, e.g. object A is on top of object B are considered.

Why not implement true visual perception instead of text descriptions?

What insights can be drawn about LLMs' capabilities for embodied planning?

The authors have clarified a few things and made improvements. I upgraded my score.

---

> ### Author Response · Authors · 2024-11-22
> **Rebuttal by Authors**
>
> We extend our gratitude to the reviewer for their insightful comments. Below, you will find our comprehensive responses to your primary questions.
> 1. Comments: Limited novelty in contributions: Task generation using LLMs has been explored extensively in prior work The benchmark tasks are relatively simple and constrained
>
> Response: We wish to further clarify this point with the reviewer. We compared our benchmark with previous work, as presented in Table 1, and demonstrated the advantages of our proposed benchmark. "Including modality (natural language, vision, or both), vocabulary type (open or constrained), data size, and the capacity for unlimited data generation via the provided pipeline. Additionally, it assesses whether benchmarks evaluate the planning abilities of large language models (LLMs), whether data generation is automated or human-annotated, and the consideration of spatial, temporal, or causal constraints." In addition, we also delineated the position of our work in Section 2 Related Work.
>
> 2. Comments: Significant limitations in the experimental setup: The simulators (Virtual Home and Habitat) are quite restricted in terms of task complexity and environmental interactions Agents do not have true perception - they receive text descriptions rather than visual input. The spatial and temporal constraints are quite basic
>
> Response: We kindly wish to clarify that there may be some misunderstandings regarding our paper. The entire LLM agent pipeline process images. In the paper, we utilize the perception modules: the perception API in VirtualHome and GroundingDINO in Habitat, to extract object information from the images. This object information is then fed into the LLM to facilitate decision-making. Consequently, we do not employ VLMs at the current stage, as VLM agents present greater challenges that we intend to explore in future work. In addition, the tasks involving the spatial and temporal constraints we have proposed currently pose significant challenges for LLMs. Although the idea is basic, much of the related work does not adequately consider it.
>
> 3. Comments: Lack of rigorous analysis and insights: The LLM agent comparison results are presented as they are without novel insights No clear framework for systematic task generation and evaluation Limited discussion of how the findings could improve embodied AI systems
>
> Response: Thanks for your comments. 1) We have incorporated insights of LLM agent comparison results in the revised version: "To verify the validity of our proposed benchmark and to demonstrate that tasks become more challenging after the addition of spatial or temporal constraints, we present comprehensive experimental results to explore the performance of our proposed LLM agent across various tasks in multiple simulators." and "To assess the impact of various LLMs on the evaluation process, we have additionally evaluated the benchmark with more LLM models." 2) We have revised our paper to the best of our ability to improve clarity of framework for systematic task generation and evaluation. 3) We have highlighted the primary contributions of our paper in the Section Conclusion: "Overall, our proposed benchmark can assist researchers in evaluating the performance of their LLM agents in completing complex embodied planning tasks that involve spatial and temporal constraints."
>
> 4. Comments: Compared to recent work like "Embodied Agent Interface: Benchmarking LLMs for Embodied Decision Making https://arxiv.org/abs/2410.07166" (note that this is not considered part of related work as the submission is concurrent), this paper: Lacks a principled framework for formalizing different types of embodied tasks Does not provide fine-grained error analysis Misses systematic evaluation of different LLM capabilities
>
> Response: In Figure 3, this framework formalizes various types of embodied tasks through a task classification module and task-specific component retrieval module. For the purpose of better evaluation, we assess each type of task to determine its difficulty. Additionally, we present Case Study and Error Analysis in Appendix A.1 Successful Cases of Embodied Planning Tasks and Appendix A.2 Failed Cases of Embodied Planning Tasks. Furthermore, we evaluated several different LLMs for our proposed benchmark, as shown in Table 3: Evaluation with open source and closed source LLMs including GPT-4, Llama3-70B, Mistral-8x7B and Llama3-8B.
>
> 5. Comments: The writing is quite poor. It did not explain the setup clearly. I have to find information in the appendix to continue reading.
>
> Response: We apologize for the extensive appendices included. Due to the page limit, we had to relocate many details to the Appendices. We have revised our paper to enhance clarity.

---

> > ### Author Response · Authors · 2024-11-22
> > **Rebuttal by Authors (second part)**
> >
> > 6. Comments: How does the task generation ensure coverage of meaningful spatial-temporal constraints? It seems that only limited constraints, e.g. object A is on top of object B are considered.
> >
> > Response: We utilized four spatial constraints between objects: on, inside, close to, and facing because Virtual Home offers limited spatial relations. These relations between objects were extracted from the Virtual Home Relation Scene Graph, ensuring that each spatial constraint is valid within the scene. To generate tasks with temporal constraints, we re-arranged objects using templates; however, some of these rearrangements could be unreasonable. Therefore, we employed verification through the Virtual Home simulator and a LLM to ensure that the temporal constraints are meaningful, as illustrated in the post-processing module in Figure 2. More details are provided in Section 3.2 Task Generation Using LLMs: LLMs assist task generation & Post-filtering based on the executability in simulator.
> >
> > 7. Comments: Why not implement true visual perception instead of text descriptions?
> >
> > Response: We wish to further clarify this point with the reviewer. We utilized the perception API in VirtualHome and GroundingDINO in Habitat to interpret visual signals. Specifically, we assessed the state-of-the-art open-vocabulary object detection model, GroundingDINO, on images from VirtualHome, with additional details available in Appendix A.10.3. Our evaluation indicated that the perception capability was suboptimal due to the inadequate image quality in VirtualHome. As a result, we employed the 'get_object_list' function from the VirtualHome Perception API, which leverages first-person perspective images.
> >
> > 8. Comments: What insights can be drawn about LLMs' capabilities for embodied planning?
> >
> > Response:  Following your suggestion, we have included these insights in Section 1 Introduction: "Embodied planning refers to an agent's ability to formulate plans and execute tasks within a physical environment. Large Language Models (LLMs) and Vision Language Models (VLMs) have demonstrated significant advancements in vision understanding, natural language comprehension, and generation. Although LLMs and VLMs are not inherently designed for embodied planning, there is potential for these models to contribute to this field. LLMs and VLMs possess an extensive repository of knowledge derived from their training data, which enables them to comprehend and generate contextually relevant advice and strategies. Additionally, they have the capability to translate complex tasks into step-by-step instructions through interactions with humans. Furthermore, they can refine plans based on feedback from the environment or human interventions."

---

> > > ### Comment · Reviewer_sWrr · 2024-11-28
> > > **thanks for the rebuttal**
> > >
> > > Thanks a lot for the rebuttal. It has clarified several comments.
> > >
> > > "Comments: What insights can be drawn about LLMs' capabilities for embodied planning?
> > > Response:  Following your suggestion, we have included these insights in Section 1 Introduction: "Embodied planning refers to an agent's ability to formulate plans and execute tasks within a physical environment. Large Language Models (LLMs) and Vision Language Models (VLMs) have demonstrated significant advancements in vision understanding, natural language comprehension, and generation. Although LLMs and VLMs are not inherently designed for embodied planning, there is potential for these models to contribute to this field. LLMs and VLMs possess an extensive repository of knowledge derived from their training data, which enables them to comprehend and generate contextually relevant advice and strategies. Additionally, they have the capability to translate complex tasks into step-by-step instructions through interactions with humans. Furthermore, they can refine plans based on feedback from the environment or human interventions."
> > >
> > > These are quite high level and well-known. Do you have more specific insights unique from your study?

---

> > > > ### Author Response · Authors · 2024-11-29
> > > > **Thanks for your feedback**
> > > >
> > > > Thank you for your feedback. In response to your request for more specific and unique insights derived from our study, we have delved deeper into the specific contributions and observations related to the capabilities of LLMs/VLMs in embodied planning. Below, we have outlined these insights:
> > > >
> > > > 1. Generalization or transferable knowledge: Our study highlights the ability of LLMs/VLMs to leverage their pre-existing knowledge in novel environments. We observed that these models can apply contextually relevant knowledge to new and unfamiliar tasks, which is advantageous in dynamic and unpredictable physical settings. Specifically, our research explored how LLMs manage real-time adaptation in response to changing environmental factors. We found that the models can adjust their strategies based on feedback. This adaptability is crucial for embodied planning in unstructured environments where conditions can change rapidly. As task complexity increases, conventional expert rule-based methods become less efficient. Therefore, the enhanced capabilities of LLMs/VLMs can progressively improve embodied planning efficiency.
> > > >
> > > > 2. Hierarchical task decomposition: A distinctive finding from our study is the competence of LLMs in decomposing complex tasks into manageable sub-tasks. This hierarchical approach enables better execution of plans, especially for multi-step processes that necessitate sequential actions. The models’ ability to break down tasks and address each step individually contributed to higher success rates in execution. Moreover, LLMs demonstrate an ability to comprehend human instructions, particularly those that cannot be parsed by some solvers easily.
> > > >
> > > > 3. Identified limitations and constraints: In our research, we have identified specific challenges faced by existing LLMs/VLMs in tasks that require planning actions within a physical environment. These challenges are particularly pronounced in their ability to understand spatial relations, i.e., the positioning of objects and their relative spatial orientations in space, and temporal relations, i.e., the unfolding of events over time. A notable example of these limitations is the difficulty that these models encounter when assessing the proximity of two objects. For instance, if the task is locating object A near object B, LLMs and VLMs often struggle to accurately interpret this spatial relationship, although it is relatively easy for humans. This difficulty arises probably because their training do not adequately support precise understanding or reasoning about physical distances and positions. Our performance evaluations and benchmarks have confirmed the need for improvements in these areas. We are optimistic that future iterations of LLMs/VLMs will demonstrate significant advancements. Specifically, we anticipate that these models will become more adept at comprehending and reasoning about the physical aspects of the world. This includes accurately understanding the locations of objects relative to one another (spatial understanding) and the sequence in which events occur (temporal understanding). In summary, our study has highlighted the current shortcomings of LLMs/VLMs in handling spatial and temporal information, underscoring a critical need for future improvements to enhance their comprehension of the physical world.
> > > >
> > > > We appreciate your suggestion, and we believe these points address the need for more specific insights derived from our research.

---

> > > > > ### Author Response · Authors · 2024-12-02
> > > > > **Looking forward to hearing from you soon**
> > > > >
> > > > > Dear reviewer,
> > > > >
> > > > > We want to follow up on our previous response and the improvements we've made based on your insightful review. As the rebuttal period nears its end, we would like to ensure that we have addressed all your concerns. Your feedback is invaluable to us, and we welcome any additional comments you might have.
> > > > >
> > > > > Thank you for your time and effort in helping us enhance our work.
> > > > >
> > > > > We are looking forward to your response.

---

### Official Review · Reviewer_bTsZ · 2024-11-04

**Soundness:** 2
**Presentation:** 2
**Contribution:** 2
**Rating:** 3
**Confidence:** 4

**Summary:**

The manuscript proposes an Embodied AI benchmark, meant to focus on assessing models' abilities to reason about spatial and temporal context. The manuscript proposes mechanisms for programmatic generation of tasks, within the VirtualHome environment, and automatic evaluation of agents — both, through the use of foundation models.

**Strengths:**

I like the setting considered by the manuscript — automatic generation and evaluation of embodied reasoning tasks.

**Weaknesses:**

Section 1 — The manuscript asserts that foundation models struggle with spatiotemporal reasoning; benchmarking Vision-Language-Action (VLA) models seems particularly relevant to this assertion.

Section 2 — The manuscript states, "We introduce a benchmark for household embodied task planning rather than question answering about the environment. It focuses on skill-level task decomposition rather than lower-level control." (L179-181). This is not sufficiently motivated. As better foundation models become available, skill-level task decomposition requires less analysis and optimisation, compared to spatiotemporal reasoning for low-level control.

Section 3 — Is there a risk that, if tasks and evaluations are performed by LLMs, methods that consist of the same LLMs may obtain inflated performance evaluations? The evaluations would then be in-domain. This phenomenon would be catastrophic, as agents' behaviors would be rewarded in ways that do not necessarily lead to improved behavior in the real-world. For validating the effectiveness of this benchmark for suitability in assessing model performance, the manuscript should establish that performance in this benchmark correlates with performance, e.g., on tasks extracted from multiple pre-existing benchmarks and/or on real-world tasks from large-scale datasets, (e.g., DROID, OXE). Furthermore, to further avoid the test-set leakage issue above, perhaps the proposed method for task generation should include some sort nondeterminism, e.g., through a set of structured perturbations, common in the safety-critical scenario generation literature.

Section 3 — Insufficient evaluation. The manuscript provides only very limited cross-simulator analysis. What would be better would be to take multiple representative model classes, identify similar cross-simulator tasks according to all the conditions that the manuscript proposes (Section 3.1), and benchmark the models in a principled way. Currently, it is difficult to extract insights about the relative usefulness of the proposed framework. Furthermore, direct comparisons/extensions with RoboGen are missing, despite the fact that RoboGen seems to share the same motivation.

Section 4 — The manuscript does not provide any new insights about *why* it asserts that foundation models have struggled with spatiotemporal reasoning. It would be useful if the manuscript could offer discussion there, perhaps through the task categorization proposed by the manuscript.

Section 4 — As new foundation models are introduced, they change sometimes-dramatically in their capabilities, and not always entirely for the better. How does the proposed framework limit the failure cases of foundation models from affecting the efficacy of, e.g., model evaluation in the proposed benchmark? It seems that the usefulness of the benchmark is directly tied to the underlying models’ abilities to perform task generation and evaluation flawlessly.

**Questions:**

No additional questions — please see above.

---

> ### Author Response · Authors · 2024-11-22
> **Rebuttal by Authors**
>
> We thank the reviewer for their insightful comments. Please find our overall response to your main questions below.
> 1. Comments: Section 1 — The manuscript asserts that foundation models struggle with spatiotemporal reasoning; benchmarking Vision-Language-Action (VLA) models seems particularly relevant to this assertion. https://arxiv.org/html/2411.05821v1
>
> Response: Thank you for sharing the paper. However, since it was posted online after the ICLR submission deadline, we were unable to review it. Upon reviewing this paper, I am uncertain where the discussion of spatial reasoning is addressed, although temporal reasoning is mentioned but unclear.
>
> 2. Comments: Section 2 — The manuscript states, "We introduce a benchmark for household embodied task planning rather than question answering about the environment. It focuses on skill-level task decomposition rather than lower-level control." (L179-181). This is not sufficiently motivated. As better foundation models become available, skill-level task decomposition requires less analysis and optimisation, compared to spatiotemporal reasoning for low-level control.
>
> Response: We kindly wish to clarify that there may be some misunderstandings regarding our paper. We revised this sentence for clarity: "We introduce a benchmark for household embodied task planning rather than question answering about the environment. This benchmark includes task descriptions and success criteria within a virtual environment. Any methods employing LLMs can be evaluated using our benchmark due to the substantial diversity in task descriptions, ensuring that LLMs will be employed for task decomposition at least once. Currently, the LLM component of the proposed LLM agent baseline focuses on skill-level task decomposition rather than lower-level control." Spatiotemporal reasoning also relies on LLMs/VLMs because our proposed LLM agent baseline also takes spatiotemporal reasoning into account. Low-level control pertains to actions such as moving to an intermediate position or manipulating a specific object. Since VirtualHome does not allow modification of these low-level controls and we can only utilize the low-level control tools provided by VirtualHome, we focus solely on high-level task decomposition while incorporating spatiotemporal reasoning.
>
> 3. Comments: Section 3 — Is there a risk that, if tasks and evaluations are performed by LLMs, methods that consist of the same LLMs may obtain inflated performance evaluations? The evaluations would then be in-domain. This phenomenon would be catastrophic, as agents' behaviors would be rewarded in ways that do not necessarily lead to improved behavior in the real-world. For validating the effectiveness of this benchmark for suitability in assessing model performance, the manuscript should establish that performance in this benchmark correlates with performance, e.g., on tasks extracted from multiple pre-existing benchmarks and/or on real-world tasks from large-scale datasets, (e.g., DROID, OXE). Furthermore, to further avoid the test-set leakage issue above, perhaps the proposed method for task generation should include some sort nondeterminism, e.g., through a set of structured perturbations, common in the safety-critical scenario generation literature.
>
> Response: In our study, tasks were generated using GPT-4 and evaluated using several different open-source and closed-source LLMs as summarized in Table 3. The performance of the Llama3-70B model is comparable to that of GPT-4. Consequently, there is no inflated performance evaluation. The primary objective of our work is to introduce an embodied planning benchmark rather than a novel LLM agent. We evaluated different LLMs to establish a LLM agent baseline, thereby verifying the validity of our benchmark. Therefore, our focus was not on evaluating LLMs on other benchmarks.

---

> > ### Author Response · Authors · 2024-11-22
> > **Rebuttal by Authors (second part)**
> >
> > 4. Comments: Section 3 — Insufficient evaluation. The manuscript provides only very limited cross-simulator analysis. What would be better would be to take multiple representative model classes, identify similar cross-simulator tasks according to all the conditions that the manuscript proposes (Section 3.1), and benchmark the models in a principled way. Currently, it is difficult to extract insights about the relative usefulness of the proposed framework. Furthermore, direct comparisons/extensions with RoboGen are missing, despite the fact that RoboGen seems to share the same motivation.
> >
> > Response: Thank you for your suggestion. Our research primarily focuses on developing an embodied planning benchmark and assessing tasks of varying complexity within the VirtualHome environment to validate the proposed benchmark. Furthermore, we have conducted evaluations in the Habitat environment to ensure that our task generation and evaluation pipelines are adaptable to different virtual environments. Given this comprehensive approach, we believe that the current evaluations are adequate for the scope of our paper. We wish to mention that we have considered the feature you mentioned regarding RoboGen. We discussed the distinctions between our work and RoboGen in Section 2 Related Work: "RoboGen employs LLMs to generate infinite tasks and scenes for its simulation engine. Our work distinguished by its focus on generating complex tasks in a controlled manner, utilizing various types of constraints to create more realistic and challenging embodied tasks."
> >
> > 5. Comments: Section 4 — The manuscript does not provide any new insights about why it asserts that foundation models have struggled with spatiotemporal reasoning. It would be useful if the manuscript could offer discussion there, perhaps through the task categorization proposed by the manuscript.
> >
> > Response: Some related works [References 1-3] introducing foundational models have struggled with spatiotemporal reasoning. We have provided a clearer discussion in Section 1 Introduction in the revision: "LLMs and VLMs face some challenges in understanding the physical world [References 1-3], including spatial understanding. Effective spatiotemporal reasoning often necessitates the integration of knowledge from multiple domains, such as physics and human behavior. These domains may not be adequately represented in the training data of existing foundational models."
> > References (Please note that these references were already included in the original version of the paper.):
> > [1] Baoxiong Jia, Ting Lei, Song-Chun Zhu, and Siyuan Huang. Egotaskqa: Understanding human tasks in egocentric videos. Advances in Neural Information Processing Systems, 35:3343–3360, 2022.
> > [2] Boyuan Chen, Zhuo Xu, Sean Kirmani, Brian Ichter, Danny Driess, Pete Florence, Dorsa Sadigh, Leonidas Guibas, and Fei Xia. Spatialvlm: Endowing vision-language models with spatial reasoning capabilities. arXiv preprint arXiv:2401.12168, 2024.
> > [3] Raghav Jain, Daivik Sojitra, Arkadeep Acharya, Sriparna Saha, Adam Jatowt, and Sandipan Dandapat. Do language models have a common sense regarding time? revisiting temporal commonsense reasoning in the era of large language models. In Proceedings of the 2023 Conference on Empirical Methods in Natural Language Processing, pp. 6750–6774, 2023.
> >
> > 6. Comments: Section 4 — As new foundation models are introduced, they change sometimes-dramatically in their capabilities, and not always entirely for the better. How does the proposed framework limit the failure cases of foundation models from affecting the efficacy of, e.g., model evaluation in the proposed benchmark? It seems that the usefulness of the benchmark is directly tied to the underlying models’ abilities to perform task generation and evaluation flawlessly.
> >
> > Response: We agree that foundation models are rapidly improving. At the current stage, at least GPT-4 performs well for task generation. We are capable of generating a set of tasks for benchmarking purposes and maintaining these tasks. Regarding the proposed evaluation framework, we introduced an evaluation baseline to verify the validity of our proposed benchmark. The LLM used in the evaluation pipeline can be any foundation model, and this pipeline can be further optimized or replaced by other researchers. We mentioned this in Section-Conclusion: "Given that we merely introduced an LLM agent as a baseline for benchmark evaluation, there remains considerable potential for baseline improvement in future research. "

---

> > > ### Author Response · Authors · 2024-12-02
> > > **Looking forward to hearing from you soon**
> > >
> > > Dear reviewer,
> > >
> > > We want to follow up on our previous response and the improvements we've made based on your insightful review. As the rebuttal period nears its end, we would like to ensure that we have addressed all your concerns. Your feedback is invaluable to us, and we welcome any additional comments you might have.
> > >
> > > Thank you for your time and effort in helping us enhance our work.
> > >
> > > We are looking forward to your response.

---

> ### Comment · Reviewer_bTsZ · 2024-12-02
>
> 1. My intention in referencing that work was not to suggest it as a single baseline that should be included; I provided the resource in the spirit of assisting the authors, so that the authors may complete their literature review and gain inspirations for tweaking their experiments to provide relevant comparisons. In the context of VLAs, OpenVLA (https://arxiv.org/pdf/2406.09246v1) is one model that comes to mind for many; that manuscript first became available in June. The reason for requesting VLA baselines was that these are a class of models that pursue better spatiotemporal reasoning for low-level control, but may still have some limitations, which seems central to verifying the manuscript's claim here. This underscores one of my main concerns with the present manuscript in that there seems to be lacking comparisons with other environments and lacking inclusion of relevant baselines.
>
> 2. It seems that there was no misunderstanding about the paper here. In my original review, I suggested that the manuscript did not sufficiently motivate why skill decomposition should be prioritized over spatiotemporal reasoning for low-level control. In the authors' response, they now explain that the reason was that the experimental environments they *selected* do not support the study of spatiotemporal reasoning for low-level control. I still feel as if this is insufficient motivation for even the revised statement. Are the authors suggesting that there are no environments that can be used for studying 'spatiotemporal reasoning for low-level control'? If so, this needs to be asserted and justified. If this is not what the manuscript intends to suggest, it should comment on why it did not select better environments for its analysis.
>
> 3. The manuscript proposes a new benchmark. For a new benchmark to be useful for the community, the manuscript must establish a few things: (a) the benchmark can be used to identify the reasons why different model types have different strengths and weaknesses; (b) the benchmark sheds light on a relatively new problem, perhaps one that is under-explored in the literature, so that it cannot be easily solved "tomorrow" and/or encourages significant methodological progress to be made; and (c) the benchmark provides valuable contributions over other similar benchmarks. The proposed benchmark mostly pursues (a), but I feel like (b) and (c) are still lacking. I discuss further concerns with (c) in the next point.
>
> 4. The authors offered high-level discussion of their work, relative to RoboGen, in Section 2 Related Work: "RoboGen employs LLMs to generate infinite tasks and scenes for its simulation engine. Our work distinguished by its focus on generating complex tasks in a controlled manner, utilizing various types of constraints to create more realistic and challenging embodied tasks." How can the proposed work claim that its tasks are "more realistic and challenging", without any form of direct comparison and without metrics that govern, e.g., 'realism' and 'environment complexity'? I (still) feel as if the manuscript is lacking crucial experiments here.
>
> 5. See point #1, above.
>
> 6. As I mentioned in my original review, "it seems that the usefulness of the benchmark is directly tied to the underlying models’ abilities to perform task generation and evaluation flawlessly." It is true that other researchers _can_ switch out and optimize the evaluating LLM agent, in order to obtain an improved evaluation of model performance under the proposed framework. In order to motivate the community to do this, however, the manuscript should demonstrate that using an LLM to evaluate model performance is **reliable**. Understanding models' behavior, detecting failure, suggesting grounded remedial actions, scoring performance, etc. are active research directions in the Embodied AI and Robotics communities (see, e.g., https://arxiv.org/pdf/2306.15724v1). These approaches still show significant limitations when attempts are made to use them for assessing agent behavior, hence my original question/concern: How does the proposed framework limit the failure cases of foundation models from affecting the efficacy of model evaluation?
>
> I am also monitoring the discussions with the other reviewers, and I still retain the above concerns from my original review.
>
> I will retain my current score for now.

---

> > ### Author Response · Authors · 2024-12-04
> >
> > (1. OpenVLA is a valuable reference, and we will include it in the literature review in our revision. However, our benchmark focuses on high-level decomposition rather than low-level control, which we will elucidate further in the subsequent point.
> >
> > (2. We acknowledge that low-level control is an integral aspect of embodied planning. Since high-level decomposition and low-level control can be decoupled in embodied planning tasks, our benchmark primarily focuses on high-level decomposition, which encompasses task-level spatial and temporal constraints. Consequently, low-level control is not prioritized in our study. Additionally, benchmarks related to low-level control have already been extensively explored, such as RoboGen, in contrast to high-level decomposition. Virtual Home serves as an excellent environment for concentrating on high-level decomposition, providing all necessary low-level controls through its API. Therefore, we chose to concentrate exclusively on high-level decomposition, without considering low-level control strategies, as these can also significantly impact overall embodied planning performance.
> >
> > (3. Regarding point (b), we have already thoroughly compared the differences between our benchmark and others in Section 2 Related Work, as well as in Table 1.
> >
> > (4. More realistic tasks refer to activities commonly performed in daily household chores, such as locating an apple and placing it in the refrigerator. These types of tasks integrate both navigation and manipulation. However, RoboGen focuses solely on manipulation tasks. RoboGen's tasks lack spatial constraints and temporal constraints. In contrast, our benchmark tasks encompass both navigation and manipulation capabilities and integrate spatial and temporal constraints. Consequently, our benchmark pose more challenging scenarios. We are uncertain about how to experimentally compare different benchmarks and evaluate their metrics in this context.
> >
> > (6. Firstly, we ensure that our generated task meets high standards of quality. Incorrect tasks are filtered through a two-step process. First, we employ LLMs to identify and exclude unreasonable or unusual tasks. The prompt used for this filtering process is detailed in Table 9 of Appendix A.12. The second step involves simulator verification. If the generated task includes at least one shortest path to locate the target object, and this path can be successfully executed in the simulator, the task is considered correct. Otherwise, it is deemed incorrect. Moreover, we employ human evaluation to ensure that the majority of the tasks we generate appear reasonable. Although a small number of tasks were deemed unreasonable or unusual based on our human evaluation, they can still be successfully executed in the Virtual Home simulator. One example of an unreasonable task by human evaluation we mentioned in the paper is "Find the face cream tube and put it into the folder." While some might consider this task reasonable or common in practice, we classify it as unusual due to our relatively strict human evaluation criteria. Other examples that were deemed unreasonable or unusual based on human evaluation are as follows: "Your task is to retrieve cellphone and deposit it into box, subsequently transferring both into garbagecan.", "Find a bunch of bananas and put them into the basket located by the toilet." and "Find the toothbrush and place it in the coffeemaker's cup." Secondly, we did not use an LLM to evaluate the model's performance; rather, we integrated an LLM as a component of the agent, referred to as the LLM agent.

---

### Official Review · Reviewer_XAtN · 2024-11-05

**Soundness:** 2
**Presentation:** 3
**Contribution:** 2
**Rating:** 5
**Confidence:** 3

**Summary:**

This paper introduces a benchmark for evaluating embodied task planning. The authors propose using LLMs to automatically generate new tasks, including their descriptions and completion criteria. A modular approach is introduced to solve the proposed task. This modular approach can be combined with different LLMs to evaluate their planning abilities.

**Strengths:**

1. This paper introduces a benchmark that can be scaled up using LLMs to automatically generate the tasks and their completion criteria.
2. Different LLM models are evaluated on the proposed tasks, and insights about the models and challenges introduced by different types of tasks are drawn.
3. The paper shows that after finetuning automatically generated data based on the tasks, the open-sourced model can perform on par with a proprietary model.

**Weaknesses:**

1. Although the paper claims that an infinite amount of tasks can be generated. It's unclear whether the diversity is bottlenecked by the scenes the simulator provides and the actions (including the transition models of the actions) the simulator supports.
2. It's unclear what new conclusions or insights can be drawn from this benchmark. I hope the authors can provide a more detailed analysis and categorize the failure modes.
3. In the proposed modular approach, there are perception modules. It's unclear whether the robustness of these perception modules is being evaluated.
4. The authors should justify the distribution of the generated tasks shown in Figure 1. It's unclear whether the distribution is directly from the generation pipeline or deliberately achieved.
5. Since the generated tasks themselves could be wrong. Are the incorrect tasks filtered? I hope the authors can provide details on how the incorrect tasks are accounted for, especially when the methods peform similarly.
6. It would be great to see qualitative examples of both positive and negative examples.
7. Some definitions of the tasks can be elaborated on: 1) it's unclear what difference between regular navigation and navigation due to the distant initial position. 2) how is the optimal path defined? 3) since there are different types of temporal and causal constraints, what is the percentage of these types in the generated tasks?

**Questions:**

I hope the authors can address my questions and comments above.

---

> ### Author Response · Authors · 2024-11-22
> **Rebuttal by Authors**
>
> We thank the reviewer for insightful comments. Please find our overall response to your main questions below.
> 1. Comments: Although the paper claims that an infinite amount of tasks can be generated. It's unclear whether the diversity is bottlenecked by the scenes the simulator provides and the actions (including the transition models of the actions) the simulator supports.
>
> Response: Thank you for your comments. Although the number of scenes (e.g. 50 scenes for VirtualHome) and actions supported by the simulator are limited, and we acknowledge the difficulty in designing new actions within the simulator, an unlimited number of scenes, objects and their layout can be easily created and loaded to virtual environments. Theoretically, this diversity of scenes and objects should be sufficient. In addition, we can also generate infinite number of tasks by changing the initial position of the agent in the same scene.
>
> 2. Comments: It's unclear what new conclusions or insights can be drawn from this benchmark. I hope the authors can provide a more detailed analysis and categorize the failure modes.
>
> Response: Thank you for your suggestion. 1) We have made revisions to the Conclusion section by adding new insights, as follows: "Overall, our proposed benchmark can assist researchers in evaluating the performance of their LLM agents in completing complex embodied planning tasks that involve spatial and temporal constraints." 2) Additionally, we included some key contributions of our paper in the Introduction section: "Embodied planning refers to an agent's ability to formulate plans and execute tasks within a physical environment. Large Language Models (LLMs) and Vision Language Models (VLMs) have demonstrated significant advancements in vision understanding, natural language comprehension, and generation. Although LLMs and VLMs are not inherently designed for embodied planning, there is potential for these models to contribute to this field. LLMs and VLMs possess an extensive repository of knowledge derived from their training data, which enables them to comprehend and generate contextually relevant advice and strategies. Additionally, they have the capability to translate complex tasks into step-by-step instructions through interactions with humans. Furthermore, they can refine plans based on feedback from the environment or human interventions. However, currently, LLMs and VLMs face some challenges in understanding the physical world, including spatial understanding. Effective spatiotemporal reasoning often necessitates the integration of knowledge from multiple domains, such as physics and human behavior. These domains may not be adequately represented in the training data of existing foundational models. To further explore this area, we introduce a new embodied task planning benchmark, ET-Plan-Bench, which features an automatic embodied task generation and evaluation pipeline that is designed to evaluate tasks with spatial and temporal understanding of the environment." 3) Furthermore, a discussion on failure cases has been introduced in the original paper: "The spatial relation constraints poses extra challenges. For example, if the task requires finding a glass near a monitor, the monitor might block the glass from the robot's view, preventing it from correctly verifying the spatial relationship. Detailed explanation can be found in Appendix A.2." 4) Due to page limitations, a detailed explanation of all failure modes can be found in Appendix A.2.
>
> 3. Comments: In the proposed modular approach, there are perception modules. It's unclear whether the robustness of these perception modules is being evaluated.
>
> Response: We wish to further clarify this point with the reviewer. We employed the perception API in VirtualHome and GroundingDINO in Habitat. We evaluated the state-of-the-art open-vocabulary object detection model GroundingDINO on VirtualHome images, with further details provided in Appendix A.10.3. Our evaluation revealed that the accuracy of GroundingDINO was suboptimal in VirtualHome due to the inadequate image quality. Consequently, we used the VirtualHome Perception API function 'get_object_list,' which is based on first-person perspective images. While this approach is not flawless, it is acceptable given that we cannot alter the API and it represents the best available option. We also addressed this limitation in Section 4.3: "Some instances of failure are attributable to defects in the perception module, which is provided by the VirtualHome API." Overall, the perception module also influences the overall performance.

---

> ### Author Response · Authors · 2024-11-22
> **Rebuttal by Authors (second part)**
>
> 4. Comments: The authors should justify the distribution of the generated tasks shown in Figure 1. It's unclear whether the distribution is directly from the generation pipeline or deliberately achieved.
>
> Response: The distribution depicted in Figure 1 is achieved incidentally rather than deliberately. Since the task can theoretically be generated infinitely, we utilized the generation pipeline shown in Figure 2 to produce a sufficient number of tasks for evaluation. After automatically filtering out unreasonable tasks, we fixed the remaining tasks for evaluation and recorded their number in Figure 1.
>
> 5. Comments: Since the generated tasks themselves could be wrong. Are the incorrect tasks filtered? I hope the authors can provide details on how the incorrect tasks are accounted for, especially when the methods perform similarly.
>
> Response: Yes, incorrect tasks are filtered through a two-step process. First, we employ LLMs to identify and exclude unreasonable or unusual tasks. The prompt used for this filtering process is detailed in Table 9 of Appendix A.12. Examples of unreasonable tasks detected through this method include: "Please put the cellphone into the bowl and put bowl on the table" and "Locate the toothbrush into condiment shaker, then place them on the floor". The second step involves simulator verification. If the generated task includes at least one shortest path to locate the target object, and this path can be successfully executed in the simulator, the task is considered correct. Otherwise, it is deemed incorrect.
>
> 6. Comments: It would be great to see qualitative examples of both positive and negative examples.
>
> Response: Due to the limitations on the main content page, we have provided positive examples in Appendix A.1 and negative examples in Appendix A.2.
>
> 7. Comments: Some definitions of the tasks can be elaborated on: 1) it's unclear what difference between regular navigation and navigation due to the distant initial position. 2) how is the optimal path defined? 3) since there are different types of temporal and causal constraints, what is the percentage of these types in the generated tasks?
>
> Response: Following your suggestion, 1) We have added clearer information regarding the definition of navigation with occlusion due to distant initial position in Section 3.1 titled Benchmark Task Definition in the revision: "We designed tasks that target objects located far from the robot's initial position. The robot is then required to locate the target object over a greater distance." In addition, in Section 4.2 Evaluation Results and Main results in Virtual Home, we have discussed the navigation with occlusion due to distant initial position: "the tasks of which the distance between the robot's initial position and the target object position is top 20% largest" 2) For the navigation and manipulation of multiple objects, we provide two versions to complete this task: one utilizing a single arm, and the other utilizing two arms. In the revision, we have included clearer information regarding the definition of the optimal path: "We consider the two-arm version to be the optimal path." 3) Theoretically, an infinite number of tasks can be generated. However, we have provided the statistics for all tasks for evaluation in Figure 1.

---

> > ### Author Response · Authors · 2024-12-02
> > **Looking forward to hearing from you soon**
> >
> > Dear reviewer,
> >
> > We want to follow up on our previous response and the improvements we've made based on your insightful review. As the rebuttal period nears its end, we would like to ensure that we have addressed all your concerns. Your feedback is invaluable to us, and we welcome any additional comments you might have.
> >
> > Thank you for your time and effort in helping us enhance our work.
> >
> > We are looking forward to your response.

---

> > > ### Comment · Reviewer_XAtN · 2024-12-02
> > >
> > > I would like to thank the authors for the detailed response. However, I still have the following concerns.
> > > 1. I agree with the authors that changing the initial configurations of the objects can result in an infinite amount of tasks, but the diversity will still be limited.
> > > 2. I think it's interesting to see examples of different failure cases, but the paper can be further strengthened by including a statistical analysis of these different failure modes and general insights drawn from these examples.
> > > 3. It would be more informative to see the effect of the perception modules on task performance since this paper aims to benchmark embodied task planning rather than symbolic task planning.
> > > 4. The definition of the task type "navigation with occlusion due to distant initial position" seems somewhat arbitrary, as it is based on the distance and the top 20th percentile. A more straightforward definition could involve directly checking whether the object is occluded by evaluating the agent's view in the simulation.

---

> > > > ### Author Response · Authors · 2024-12-04
> > > > **Thanks for your response**
> > > >
> > > > We appreciate the reviewer's additional comments. Please find our responses to your questions below.
> > > >
> > > > 1. Comments: I agree with the authors that changing the initial configurations of the objects can result in an infinite amount of tasks, but the diversity will still be limited.
> > > >
> > > > Response: Thank you for your comment. While it is true that the number of tasks can be infinite, there may indeed be a theoretical limit to their diversity. However, we assert that the diversity of tasks in our framework is both sufficient and significantly richer compared to other existing benchmarks. We will update this description in the revised version of the paper.
> > > >
> > > > 2. Comments: I think it's interesting to see examples of different failure cases, but the paper can be further strengthened by including a statistical analysis of these different failure modes and general insights drawn from these examples.
> > > >
> > > > Response: Thanks for your suggestion, we apologize for the absence of a quantitative statistical analysis of the various failure modes at this time. Our findings indicate that failures predominantly occur due to the agent reaching its maximum number of steps. This issue appears to be related either to the detection modules or to incorrect direction exploration. It is challenging to quantify the failure modes, not only because Virtual Home does not provide detection ground truth but also due to the difficulty in determining the correctness of an intermediate step until the final target object is identified. An individual step may appear locally sub-optimal but could contribute to a globally optimal outcome, or conversely, a locally optimal step might lead to a globally sub-optimal result. Each instance of failure can be attributed to multiple factors.
> > > >
> > > > 3. Comments: It would be more informative to see the effect of the perception modules on task performance since this paper aims to benchmark embodied task planning rather than symbolic task planning.
> > > >
> > > > Response: Thanks for your suggestion, but for end-to-end embodied planning tasks, it is challenging to quantitatively evaluate how perception modules affect final performance without ground-truth perception. We used the Virtual Home perception API as our perception module. Although it is not 100% accurate, alternative options are limited. For more details, please refer to Appendix A.10.3. If the target object is not detected by the perception module, the agent will continue to explore until it reaches the maximum number of steps. There is a method to manually evaluate how the perception module affects the final performance. To verify the presence of the target object in each frame, we must manually inspect the current view. This process is labor-intensive and could be subjective due to image quality.
> > > >
> > > > 4. Comment: The definition of the task type "navigation with occlusion due to distant initial position" seems somewhat arbitrary, as it is based on the distance and the top 20th percentile. A more straightforward definition could involve directly checking whether the object is occluded by evaluating the agent's view in the simulation.
> > > >
> > > > Response: Thanks for your suggestion. To avoid any misunderstandings, we will revise the definition to "navigation with a longer distance between the initial position and the target object." Whether the object is occluded cannot be predetermined in the task description because it is highly dependent on the robot's moving trajectory. Therefore, we propose that the likelihood of a target object becoming occluded increases with the distance between robot's initial position and the target object location.

---

### Meta-Review · Area_Chair_zhAq · 2024-12-18

**Metareview:**

The paper received rejection ratings from all the reviewers (5,5,5,3). The reviewers initially raised various concerns such as lack of clarity about the diversity of the tasks, insufficient motivation, insufficient evaluation, limitations of the experimental setup, and unclear presentation. The authors provided a rebuttal, but it did not change the reviewers’ opinion about the paper. The AC checked the paper, the reviews, and the responses. The AC is in agreement with the reviewers and recommends rejection.

**Additional Comments On Reviewer Discussion:**

The reviewers engaged in a detailed discussion with the authors but remained unconvinced by their responses. To highlight a few concerns: Reviewer XAtN continues to question the diversity of the tasks and the analysis of the perception module; Reviewer bTsZ is concerned about the lack of comparisons with other environments and relevant baselines; Reviewer sWrr finds the novelty limited; and Reviewer VG5n is concerned about the clarity of the paper. Hence, the manuscript does not meet the bar for publication at ICLR.

---

### Decision · Program_Chairs · 2025-01-22

Reject